# Novel pathogen introduction triggers rapid evolution in animal social movement strategies

Pratik Rajan Gupte[1]*[†], Gregory F Albery[2,3], Jakob Gismann[1], Amy Sweeny[4], Franz J Weissing[1]*

[1]Groningen Institute for Evolutionary Life Sciences, University of Groningen, Groningen, Netherlands; [2]Georgetown University, Washington, United States; [3]Wissenschaftskolleg zu Berlin, Berlin, Germany; [4]Institute of Evolutionary Biology, University of Edinburgh, Edinburgh, United Kingdom

**Abstract** Animal sociality emerges from individual decisions on how to balance the costs and benefits of being sociable. Novel pathogens introduced into wildlife populations should increase the costs of sociality, selecting against gregariousness. Using an individual-based model that captures essential features of pathogen transmission among social hosts, we show how novel pathogen introduction provokes the rapid evolutionary emergence and coexistence of distinct social movement strategies. These strategies differ in how they trade the benefits of social information against the risk of infection. Overall, pathogen-risk-adapted populations move more and have fewer associations with other individuals than their pathogen-risk-naive ancestors, reducing disease spread. Host evolution to be less social can be sufficient to cause a pathogen to be eliminated from a population, which is followed by a rapid recovery in social tendency. Our conceptual model is broadly applicable to a wide range of potential host–pathogen introductions and offers initial predictions for the eco-evolutionary consequences of wildlife pathogen spillover scenarios and a template for the development of theory in the ecology and evolution of animals' movement decisions.

*For correspondence:
pratikgupte16@gmail.com (PRG);
f.j.weissing@rug.nl (FJW)

Present address: [†]Centre for Mathematical Modelling of Infectious Diseases, London School of Hygiene and Tropical Medicine, London, United Kingdom

Competing interest: The authors declare that no competing interests exist.

## Editor's evaluation

This study provides important new insights into the effects that disease can have on movement strategies in animals. The theoretical model that forms part of this contribution generates useful predictions that are widely applicable. In doing so, it will have a lasting impact on the field.

## Introduction

Animal sociality emerges from individual decisions that balance the benefits of associations against the costs of proximity or interactions with neighbours (*Tanner and Jackson, 2012*; *Webber and Vander Wal, 2018*; *Webber et al., 2023*; *Gil et al., 2018*). Such associations can yield useful social information whether inadvertently or deliberately transmitted about resource availability (*Danchin et al., 2004*; *Dall et al., 2005*; *Gil et al., 2018*), but they also provide opportunities for the transmission of parasites and infectious pathogens among associating individuals (*Weinstein et al., 2018*; *Romano et al., 2020*; *Albery et al., 2021a*; *Cantor et al., 2021*; *Romano et al., 2022*). Wildlife pathogen outbreaks affect most animal taxa, including mammals (*Blehert et al., 2009*; *Fereidouni et al., 2019*; *Chandler et al., 2021*; *Kuchipudi et al., 2022*), birds (*Wille and Barr, 2022*), amphibians (*Scheele et al., 2020*), and social insects (*Goulson et al., 2015*). Weighing the potential risk of infection from social interactions against the benefits of social movements where to move in relation

to other individuals' positions is thus a common behavioural context shared by many animal species. Movement decisions incorporating social information the presence and status of neighbours can facilitate or reduce spatial associations and help animals balance the costs and benefits of sociality (*Albery et al., 2021a*; *Gil et al., 2018*; *Webber and Vander Wal, 2018*; *Webber et al., 2023*). Animals' social movements link landscape spatial structure, individual distributions, and the emergent structure of animal societies (*Gil et al., 2018*; *Webber et al., 2023*; *Kurvers et al., 2014*). Together, they influence the dynamics of disease outbreaks in animal populations (*White et al., 2018b*; *Romano et al., 2020*; *Romano et al., 2022*; *Keeling et al., 2001*), and such outbreaks may in turn have cascading effects on landscape structure and community ecology (*Monk et al., 2022*).

Over relatively brief ecological timescales of a few months or years, animal pathogen outbreaks can reduce social interactions among individuals due to a combination of factors. For instance, mortality from the disease may induce decreases in population density (e.g. *Fereidouni et al., 2019*; *Monk et al., 2022*), leading to fewer associations. Furthermore, adaptive behavioural responses by which animals identify infected individuals (and indeed, whether they are themselves infected) can trigger quarantining or self-isolation behaviours that reduce encounters between infected and healthy individuals overall (*Stroeymeyt et al., 2018*; *Pusceddu et al., 2021*; *Stockmaier et al., 2021*; *Weinstein et al., 2018*). When pathogens are first introduced into a population, such as during novel cross-species spillover (*Kuchipudi et al., 2022*; *Chandler et al., 2021*), fine-tuned avoidance responses are less likely, as individuals may have no prior experience of cues that indicate infection (*Weinstein et al., 2018*; *Stockmaier et al., 2021*; although general cues of infection may still play a role; see *Townsend et al., 2020*). A novel pathogen spreading through host–host contacts and imposing costs upon infected individuals could thus confer an evolutionary advantage upon less social individuals if these are also less frequently infected. Therefore, it is a common expectation that pathogen introduction broadly selects against host social behaviour, and hence against social connectivity itself (*Altizer et al., 2003*; *Cantor et al., 2021*; *Romano et al., 2022*; *Poulin and Filion, 2021*; *Ashby and Farine, 2022*).

Important aspects of animal ecology, including the transmission of foraging tactics (*Klump et al., 2021*) and migration routes (*Jesmer et al., 2018*; *Guttal and Couzin, 2010*), depend on social interactions. This makes it important to understand the long-term, evolutionary consequences of pathogen introductions for animal sociality. Climate change is only expected to make novel pathogen introductions more common (*Sanderson and Alexander, 2020*; *Carlson et al., 2022a*), making such studies more urgent. Despite this salience, novel pathogen introductions are primarily studied for their immediate demographic (*Fey et al., 2015*), and potential medical (*Wille and Barr, 2022*; *Chandler et al., 2021*; *Kuchipudi et al., 2022*; *Levi et al., 2012*) and economic implications (*Keeling et al., 2001*; *Goulson et al., 2015*; *Jolles et al., 2021*). Indeed, most introductions of novel pathogens into wildlife only come to light when they result in mass mortality events (*Fey et al., 2015*; *Wille and Barr, 2022*). Host evolutionary dynamics (and especially changes in sociality) are mostly ignored, and this is presumably because the evolution of pathogen host traits, and moreover complex behavioural traits such as sociality, is expected to be slow and not immediately relevant for management.

Theory suggests that animal sociality evolves to balance the value of social associations against the risk of pathogen transmission (*Bonds et al., 2005*; *Prado et al., 2009*; *Ashby and Farine, 2022*). However, analytical models often reduce animal sociality to single parameters, while it actually emerges from individual decisions conditioned on multiple internal and external cues. Social decision-making and movement often also vary among individuals (*Tanner and Jackson, 2012*; *Wolf and Weissing, 2012*; *Spiegel et al., 2017*; *Gartland et al., 2022*), but analytical models are unable to include individual differences in sociability. Epidemiological models based on contact networks can incorporate individual variation in social behaviour by linking these differences to positions in a social network (*White et al., 2017*; *Albery et al., 2021a*; *Albery et al., 2021b*). Yet network models often cannot capture fine-scale feedbacks between individuals' social and spatial positions (*Albery et al., 2021a*; *Albery et al., 2021b*), nor spatial variation in infection risk (*Albery et al., 2022*). Networks constructed from relatively low-resolution spatial relocation data (such as infrequent direct observations; see e.g. *Albery et al., 2021b*), may be sensitive to the network formation process when seeking to understand the rapid spread of diseases, especially if transmission has a non-linear relationship with association strength (*Farine, 2017*; *White et al., 2017*). While high-resolution animal tracking could help construct more detailed networks on which to run disease outbreak models (*Wilber et al.,*

*2022*; *Nathan et al., 2022*), such networks could also be biased by individual variation in social traits (*Gartland et al., 2022*), such as when sociality is correlated with capture probability (see e.g. *Carter et al., 2012*). Consequently, adding an explicit spatial setting to movement-disease models can be valuable in gaining a more general understanding of the interplay between social decisions, movement, and pathogen transmission (*He et al., 2021*; *Scherer et al., 2020*; *White et al., 2018c*; *White et al., 2017*).

Mechanistic, individual-based simulation models (IBMs) suggest themselves as a natural solution. IBMs can incorporate substantial ecological detail, including explicit spatial settings (*DeAngelis and Diaz, 2019*), and detailed disease transmission dynamics (*White et al., 2018a*; *Scherer et al., 2020*; *Lunn et al., 2021*; *White et al., 2018c*). Most importantly, IBMs can include individual decision-making, allowing ecological and epidemiological outcomes to emerge from individuals' movement choices. Individual-based models hitherto have focused on immediate epidemiological outcomes, such as infection persistence, and do not have an evolutionary component examining long-term consequences for either pathogens or their hosts (*White et al., 2018a*; *Scherer et al., 2020*; *Lunn et al., 2021*). Incorporating an evolutionary component to movement-disease IBMs could allow predictions on important feedbacks between the proximate ecological outcomes of infectious disease and the ultimate consequences for the evolution of host behaviour (*Cantor et al., 2021*). This could include the emergence of individual differences in the trade-offs between the costs and benefits of sociality (*Gartland et al., 2022*), with cascading effects for landscape ecology and the structure of animal societies (*Monk et al., 2022*; *Spiegel et al., 2017*; *Tanner and Jackson, 2012*; *Webber et al., 2023*). The range of animal taxa at risk from a wide array of pathogens and parasites (*Carlson et al., 2022a*; *Sanderson and Alexander, 2020*) makes it important to conceive, as a starting point, of models that can capture the key features of diverse host–pathogen dynamics and offer broad conceptual insights (*White et al., 2018c*; *White et al., 2018a*).

We built a model that seeks to capture the essential elements of animal movement decisions in the context of foraging on patchily distributed resources, under the risk of pathogen (or parasite) transmission. Our model adopts a step-selection framework in an explicit spatial setting (*Fortin et al., 2005*), allowing individuals to choose their movement directions a key component of animal movement ecology (*Nathan et al., 2008*) based on their perception of local environmental cues. These are the presence of resources (personal information) and the presence of other individuals (social information). Our model also adds an evolutionary component by allowing individuals' ecological performance (energy) over their lifetime to influence the mixture of movement strategies in their offspring generation. We examined the ecological and evolutionary consequences of the introduction of a pathogen into a novel host population (such as during cross-species spillover: *Blehert et al., 2009*; *Bastos et al., 2000*; *Wille and Barr, 2022*; *Fereidouni et al., 2019*; *Scheele et al., 2020*; *Sanderson and Alexander, 2020*; *Carlson et al., 2022a*; *Kuchipudi et al., 2022*; *Monk et al., 2022*). We modelled two scenarios of the introduction of an infectious pathogen to populations that had already evolved foraging movement strategies in its absence. Our model scenarios could be conceived as abstract representations of, among others, cross-species introductions of foot-and-mouth disease from buffalo to impala (*Bastos et al., 2000*; *Vosloo et al., 2009*), or of sarcoptic mange from llamas to vicuas (*Monk et al., 2022*), the current and historic spread of avian influenza among bird species (and more recently, spillovers into certain mammal species; *Global Consortium for H5N8 and Related Influenza Viruses, 2016*; *Wille and Barr, 2022*), of the spread of borrelliosis in novel populations of its wildlife hosts (*Levi et al., 2012*), or of SARS-CoV-2 from humans to deer (*Chandler et al., 2021*; *Kuchipudi et al., 2022*).

In scenario 1, we repeatedly introduced an infectious pathogen to a small proportion of individuals in each generation, allowing it to spread with a low probability among proximate individuals thereafter. This scenario parallels conditions that we expect are common but poorly known: that animal populations suffer pathogen introductions regularly from external sources such as individuals from an infected subpopulation of a metapopulation or sympatric heterospecifics such as those sharing breeding or wintering grounds both of these appear to be plausible events in the spread of diseases such as highly pathogenic avian influenza (*Wille and Barr, 2022*; *Global Consortium for H5N8 and Related Influenza Viruses, 2016*). We classified individuals across the evolutionary timescale of our simulation based on their inherited preferences (or selection coefficients) for environmental cues into movement strategies (similar to; ; *Bastille-Rousseau and Wittemyer, 2019*;

see 'Methods'). We compared how social information was used in movement strategies evolved before and after pathogen introductions began, and the ecological outcomes for individual movement and associations with other foragers. In a further scenario 2, we modelled only a single introduction event, but allowed the pathogen to be transmitted from parents to their offspring at the end of each generation (vertical transmission in a general sense), in addition to spreading among proximate individuals within each generation. Empirical examples of such parent-to-offspring transmission are less well known, but are implicated in the maintenance of foot-and-mouth disease in African buffalo (*Jolles et al., 2021*) and of mange among wolves (*Almberg et al., 2015*). We examined how these simulated outbreaks persisted across generations, the resulting evolutionary change in social movement strategies, and the consequences for individual behavioural outcomes. Using network epidemiological models (*Wilber et al., 2022*; *Stroeymeyt et al., 2018*; *White et al., 2017*; *Bailey, 1975*), we examined whether the spread of infections was reduced in pathogen-risk-adapted populations compared to their pathogen-risk-naive ancestors. We also investigated the effect of landscape productivity and the cost of infection, which are both expected to influence the selection imposed by pathogen transmission (*Ezenwa et al., 2016*; *Almberg et al., 2015*; *Hutchings et al., 2000*). Overall, we provide a theoretical framework applicable to a broad range of novel host–pathogen introduction scenarios and demonstrate the importance of including evolutionary dynamics in movement-disease models.

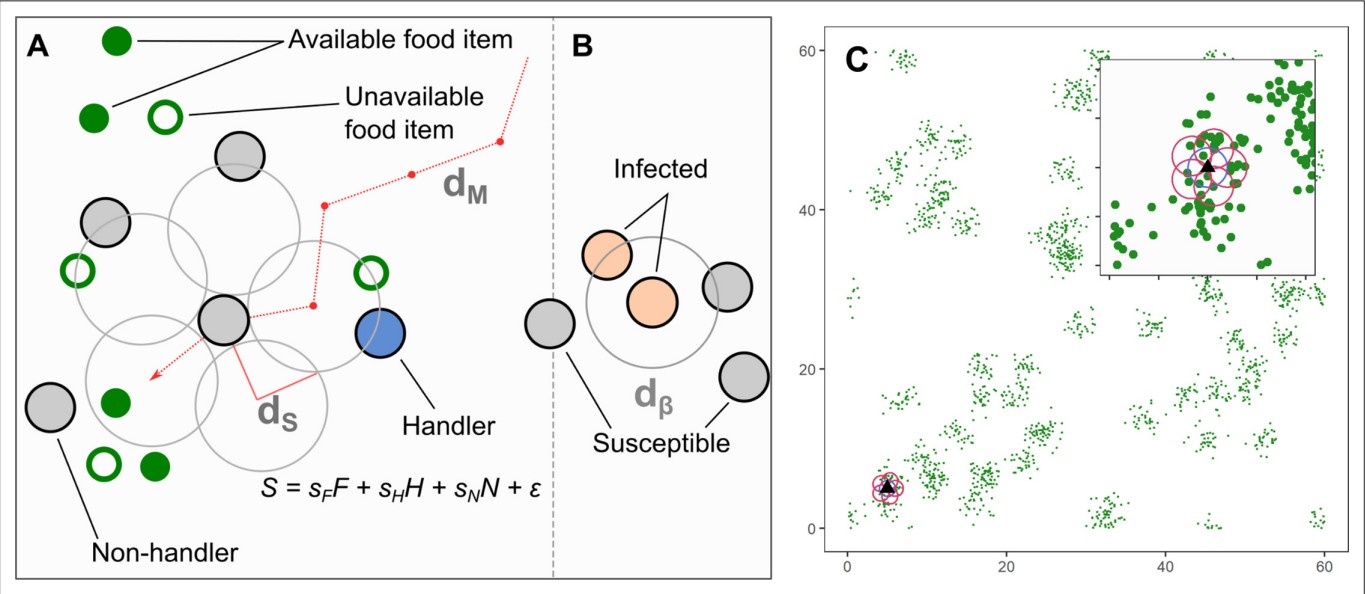

**Figure 1.** Model implementation of discrete movement steps on a landscape with continuous space, with movement steps selected based on inherited preferences for environmental cues. (**A**) Individuals search for clusters of food items (green circles), which may be immediately available (filled green circles; $F$), or may be available only in the future (open green circles). Individuals can sense only available items, and not unavailable ones. Individuals can sense other foraging individuals, and whether they have successfully found and are handling a food item (handlers; blue circles; $H$), or whether they are unsuccessful foragers still searching for food (non-handlers; filled grey circles; $N$). To decide where to move, individuals sample their environment for these cues at five locations around themselves (large open grey circles) and have a sensory range of $d_S$ (default = 1.0 units). Individuals assign each potential direction a *suitability*, $S = s_F F + s_H H + s_N N + \epsilon$, where the coefficients $s_F, s_H, s_N$ are inherited preferences for environmental cues, and $\epsilon$ is a small error term that helps break ties between locations. The sensory distance ($d_S$) and the movement distance ($d_M$) are the same, 1.0 units. (**B**) An infectious pathogen is transmitted between infected (orange circles) and susceptible (filled grey circles) individuals, with a probability $p = 0.05$, when they are within a distance $d_\beta$ of each other. In our implementation, $d_\beta$ is the same as $d_S, d_M = 1.0$ units. (**C**) An example of the resource landscape used in our simulations, consisting of 60 randomly distributed clusters of food items, with 1800 discrete food items divided among the clusters (30 items per cluster). The landscape is a square of 60 units per side, with wrapped boundaries (i.e. a torus). The food item density is 0.5 food items per unit area. Items are distributed around the centre of each cluster, within a standard deviation of 1.0 unit. Items, once consumed by foragers, are unavailable for a fixed number of timesteps (the regeneration time $R$, expressed in terms of the foragers generation time), after which they regenerate in the same location. While regenerating (i.e. unavailable), items cannot be perceived by foragers. The sensory ranges of individuals ($d_S$) are shown for each potential step (red circles, including the current location: blue circle). Food item clustering means that available items, as well as foragers handling a food item (handlers), are good indicators of the location of a resource cluster.

## Results

In our model, individuals move and forage on a landscape with patchily distributed food items, and select where next to move in their vicinity, based on inherited preferences for environmental cues food items, and other individuals (*Figure 1*). Food items, once consumed, regenerate at a rate $R$, and pathogen infection imposes a per-timestep cost $\delta E$. We classified individuals' social movement strategies in our model using a simplified behavioural hypervolume approach (*Bastille-Rousseau and Wittemyer, 2019*) based on the sign of their preferences for successful foragers handling a food item (handlers, preference $s_H$) and for unsuccessful foragers still searching for food (non-handlers, preference $s_N$).

In our models' default implementation of scenario 1, $R = 2$, food regenerates twice per generation, and $\delta E = 0.25$, that is, consuming one food item offsets four timesteps of infection. Over the 500

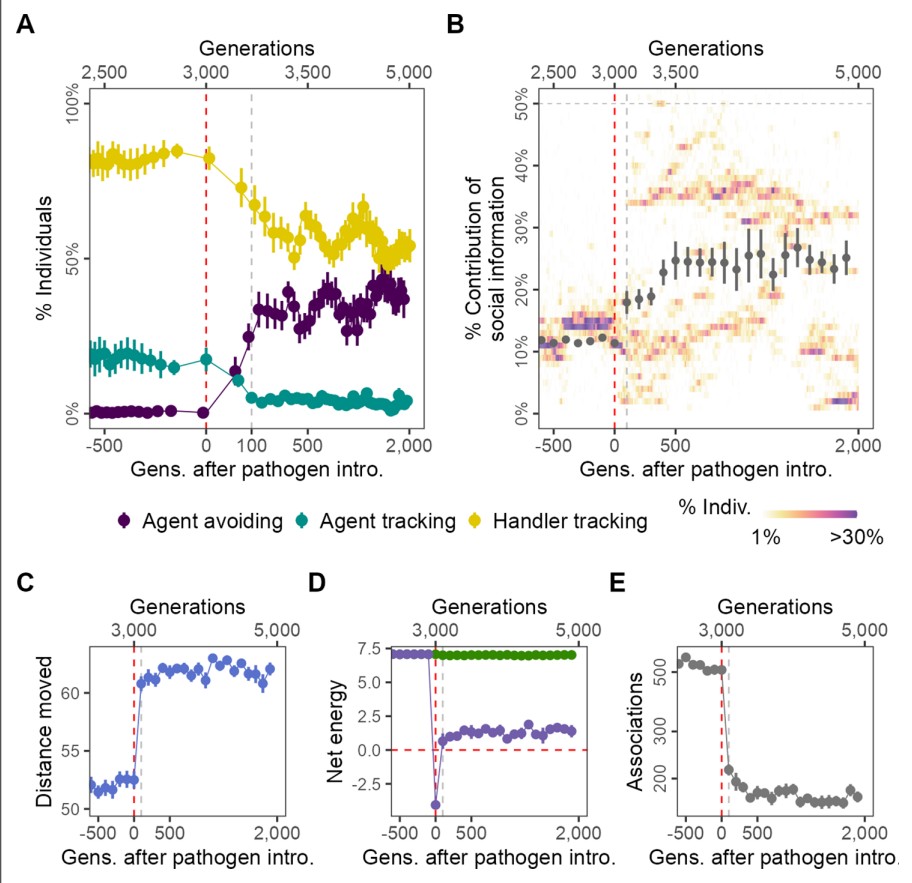

**Figure 2.** Pathogen introduction leads to rapid evolutionary changes in social information use, with cascading effects on population-level behaviour. (**A**) Before pathogen introduction in the default scenario ($R = 2$, $\delta E = 0.25$), populations rapidly evolve to mostly track handlers and avoid non-handlers (handler-tracking; $G \leq 3000$); however, the preference for food items ($s_F$) is the major determinant of their fine-scale movement decisions. Pathogen introduction leads to a rapid increase in agent avoidance which stably coexists with the handler-tracking strategy in an eco-evolutionary equilibrium. (**B**) After pathogen introduction ($G > 3000$), the importance of social cues (the presence of other individuals; the sum of the absolute, normalised preferences $s_H$, $s_N$) doubles on average (grey points; from 10% to >20%). Additionally, there is significant variation in the importance of social cues to individuals (shaded regions), which is not captured by the mean or standard error. The rapid change in social movement strategies following pathogen introduction has cascading effects on population-level behaviour. Individuals, which have evolved aversions to some kinds of foragers (depending on their strategy), (**C**) move 15% more on average, (**D**) have substantially reduced per-capita energy (purple) due to the cost of infection, as mean per-capita intake remains unchanged (green), and (**E**) also have fivefold fewer associations with other foragers. All panels show data averaged over 10 replicates, with mean and standard error; shaded regions in panel (**B**) are from a single replicate for clarity. Panels X axes begin at G = 2500,, and panel (**A**) X-axis is transformed to show the generations after introduction more clearly.

generations before the introduction of the pathogen, populations reached an eco-evolutionary equilibrium where the most common social movement strategy was to prefer moving towards handlers while avoiding non-handlers (handler-tracking; $s_H > 0, s_N < 0$) (*Figure 2A*). This is consistent with observations from a different simulation model which shares many mechanisms with this one (*Gupte et al., 2023*). A small proportion of individuals prefer to move towards both handlers and non-handlers, and are thus indiscriminately social (agent-tracking; $s_H, s_N > 0$).

## Rapid evolutionary shift in social movement strategies following pathogen introduction

Introducing an infectious pathogen to 4% (n = 20) of individuals in each generation (after G = 3000) leads to a rapid evolutionary shift that is complete within only 100 generations of pathogen introduction in how social information is incorporated into individual movement strategies. A third strategy increases in frequency: avoiding both handlers and non-handlers (agent-avoiding; $s_H, s_N < 0$; *Figure 2A*). The frequency of agent-avoiding and handler-tracking strategies is comparable within 500 generations, and fluctuates thereafter, with increases in one strategy corresponding to decreases in the other. This appears to be a dynamic equilibrium that is maintained until the end of the simulation (2000 generations after pathogen introduction; *Figure 2A*). The frequency of the agent-tracking strategy is further reduced, but the strategy never truly goes extinct, possibly due to mutations that shift $s_N$ coefficients to positive during reproduction. The section 'Effect of modelling choices on simulation outcomes' shows how the occurrence of rapid evolutionary shifts is broadly robust to modelling assumptions; in brief, such shifts also occur when (1) the pathogen reduces foraging efficiency rather than imposing a direct cost on individual energy, (2) when individuals cannot benefit from evolved adaptation to local conditions due to large-scale natal dispersal (*Badyaev and Uller, 2009*), and when (3) individuals can only reproduce if they have a positive energy balance. Furthermore, (4) evolutionary transitions away from sociality are also observed at higher but not lower handling times (a proxy for the availability of social information), and (5) both when the spatial structure of the landscape is substantially more uniform, and more clustered.

In addition to qualitative changes in social movement strategies, pathogen introduction also leads to social information becoming more important to movement decisions. Prior to pathogen introduction (*G* < 3000), individuals' handler- and non-handler preferences have only a small influence on their movement strategies ($|s_H| + |s_N|$; taken together, the contribution of social information; *Figure 2B*). Individual movement is instead guided primarily by the preference for food items ($s_F$; see 'Model output and analysis'). After pathogen introduction, there is an increase in the average importance of individuals' preferences (or aversions) for the presence of other foragers, that is, the importance of social cues (*Figure 2B*). Additionally, there is significant variation among individuals in the importance of social cues to their movement strategies, with distinct evolved polymorphisms that vary substantially between simulation replicates (*Figure 2B*). This means that the populations' mean importance of social cues does not adequately capture that some individuals assign much more importance to social cues than others, and that these distinct morphs persist in the population for many hundreds of generations after pathogen introduction.

## Population-level behavioural change due to evolutionary shift in social movement strategies

The evolutionary shift in social movement strategies causes a drastic change in population-level behaviour and outcomes (*Figure 2C–E*). There is a sharp increase in the mean distance moved by individuals; while pre-introduction individuals moved 52% of their lifetimes on average, post-introduction, individuals move for about 62% of their lifetimes (*Figure 2C*). The handler-tracking and agent-avoiding strategies lead individuals to move away from groups of individuals, with the effect of group composition on fine-scale movement decisions (handlers or non-handlers) determined by the individuals' strategy. Individuals are most likely to be found near resource clusters, and this leads to movement away from productive areas of the landscape where individuals, having acquired a food item and become immobilised, may have inadvertent associations with other foragers. Surprisingly, this does not lead to a reduction in mean per-capita intake (*Figure 2D*, green), but there is a sharp drop in mean per-capita energy (intake – total infection cost) due to the cost of infection (*Figure 2D*, purple). While strongly negative on average in the first few generations after introduction, net energy returns

to a small positive value within 100 generations of pathogen introduction. The emergence of avoidant strategies leads to a fivefold drop in encounters between individuals after pathogen introduction (*Figure 2E*), which suggests that most encounters were indeed likely taking place on or near resource clusters. These results show how even a non-fatal pathogen, by influencing the evolution of movement strategies, can have substantial indirect effects on population-level spatial and social behaviour.

## Movement-intake-sociality trade-offs and the coexistence of social movement strategies

At eco-evolutionary equilibrium in our default implementation of scenario 1 (3000 ≤ *G* ≤ 3500), the three main social movement strategies coexist, allowing a comparison of ecological and behavioural outcomes that illustrates the trade-offs between sociality, movement, and infection, which are

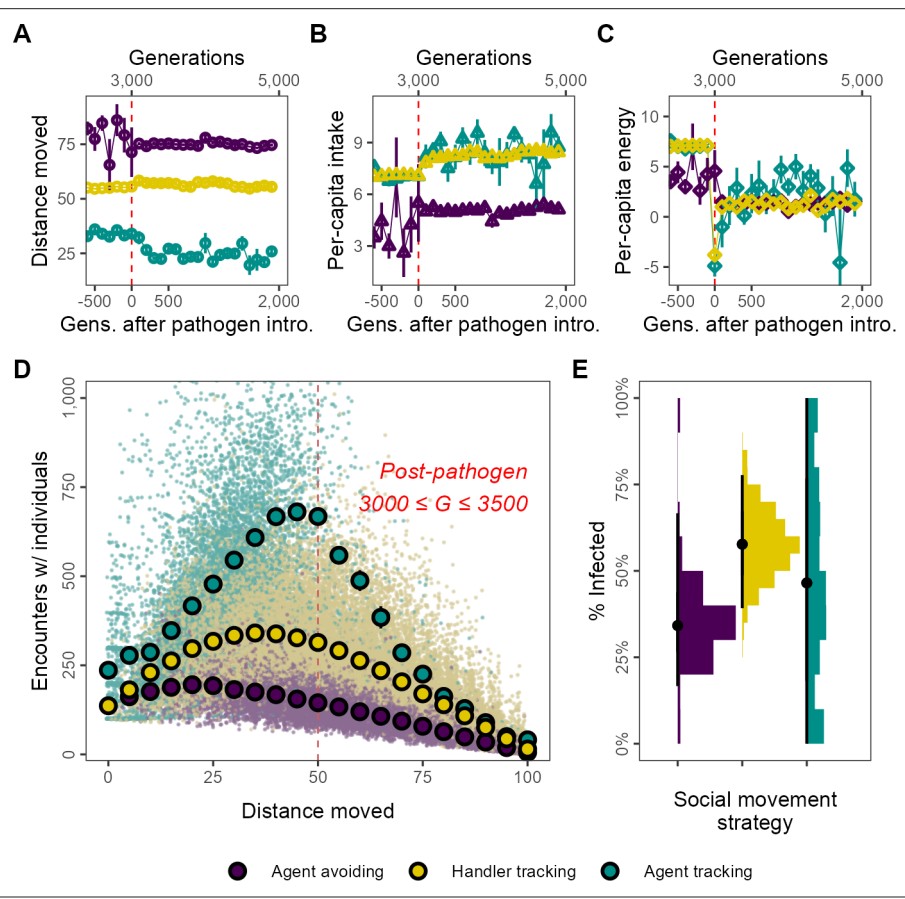

**Figure 3.** Social movement strategies coexist by trading movement for associations through dynamic social distancing, leading to differences in intake and infection rates. Population-level outcomes mask substantial variation in strategy-specific behaviour and outcomes. The three main movement strategies differ in the mean distance moved, with the agent-avoiding strategy moving substantially more (**A**), and having less intake (**B**) than the other strategies. Nonetheless, all three strategies have similar net energy and hence equivalent fitness (**C**). In post-introduction populations (3000 ≤ *G* ≤ 3500), (**D**) the distance moved by individuals of the three main strategies has a non-linear relationship with the number of associations. Individuals that move either very little (<15) or constantly (>75) have few associations. However, individuals that move intermediate distances have more associations. This curve is influenced by the social movement strategy, with agent-tracking individuals having more associations than the handler-tracking strategy for the same distance moved, while handler-tracking individuals have similarly more associations than agent-avoiding individuals. (**E**) Avoiding all other foragers leads to lower infection rates than tracking successful foragers (and avoiding unsuccessful ones; handler-tracking). Surprisingly, rare pre-introduction strategies such as following any nearby individuals (agent-tracking) may also have low infection rates, potentially due to their rarity. Panel (**D**) shows the mean and standard error for movement distance bins of five units (note standard error is very small in some cases); panel (**B**) shows infection rates; all data represent generation- and replicate-specific means (*R* = 2, *δE* = 0.25).

otherwise masked by a population-level analysis. For example, the population-level increase in movement after pathogen introduction is shown to be due to the increase in frequency of the agent-avoiding strategy as these individuals move more than handler-tracking or agent-tracking foragers (*Figure 3A*). Simultaneously, agent-avoiding individuals have a lower intake than either handler-tracking or agent-tracking individuals which have similar intakes (*Figure 3B*). Surprisingly, the more social strategies appear to increase their intake slightly following pathogen introduction this could be because exploitation competition may be reduced as agent-avoiding foragers also avoid resource clusters and have less intake than the pre-introduction average. Despite this, all three strategies have comparable if not identical net energy and hence equivalent fitness this is to be expected given their coexistence (*Figure 3C*).

The energy equivalence of the three strategies despite different per-capita intake can be explained by differing infection rates. These are in turn likely influenced by the non-linear relationship between movement and the mean number of per-capita associations of each strategy. The shape of the movement–association curve is broadly a quadratic one (*Figure 3D*). Across strategies, individuals that move more have more associations until a threshold, with associations declining from their peak as individual movement increases further; the peak of the curve is different for each strategy. For example, agent-tracking individuals that move 50 units have around 600 associations with other foragers, while

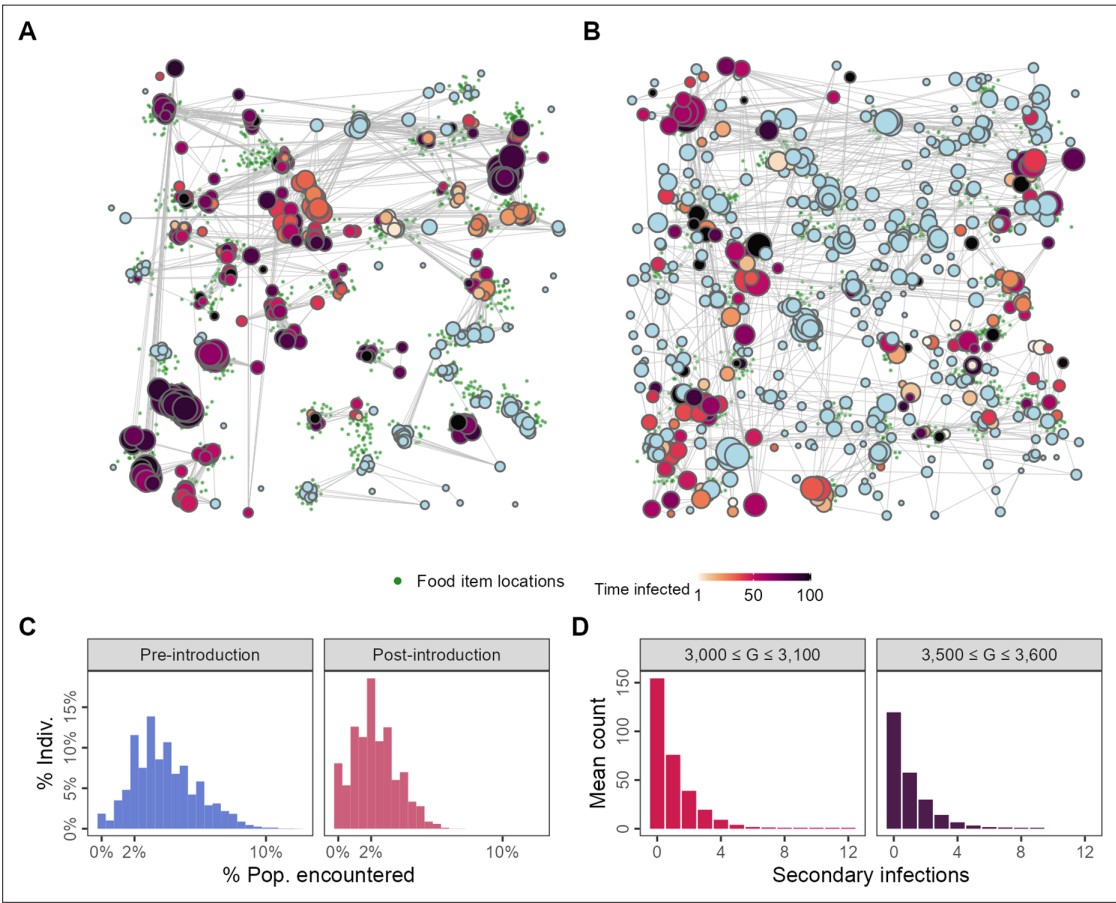

**Figure 4.** Changes to spatial-social structure in populations adapted to the presence of an infectious pathogen. Pathogen-risk-naive populations (**A**; *G* = 3000) are clustered into modules by the end of their lives, while pathogen-risk-adapted populations (**B**; *G* = 3500) are more widely dispersed over the landscape. Pre-introduction individuals encounter somewhat more unique neighbours (**C**, blue) than pathogen-risk-adapted individuals (**C**; red). (**D**) The distribution of the individual reproductive number $\nu$ is left-skewed, with most infections not resulting in any secondary cases, but has a long right-hand tail, suggesting that a small number of infected individuals are responsible for a large number of infections, suggesting that superspreading emerges from the spatial-social dynamics encoded in the model. Panels (**A**) and (**B**) show social networks from a single replicate of the default implementation of scenario 1 (*R* = 2, $\delta E$ = 0.25), while all other panels show the average of 10 replicates. Nodes represent individuals positioned at their final location in (**A**) and (**B**). Connections represent pairwise encounters (connections with weights <33rd percentile are removed for ease of visualisation), and node size represents individuals' social associations (larger = more associations). Darker node colours indicate longer infection (light blue = no infection).

handler trackers have approximately 300 associations, and agent-avoiding individuals have about 150 associations. At the extremes of movement behaviour individuals that move throughout their lifetime (movement >75) and which do not move at all (movement <15), all three strategies have similar numbers of per-capita associations; individuals that move constantly (movement = 100) have almost no associations at all. These differences likely explain why agent-avoiding and handler-tracking individuals have differing mean infection rates, at ~25 and ~33%, respectively (*Figure 3E*). Individuals of the agent-tracking strategy, on the other hand, have a wide range of infection rates (*Figure 3E*), potentially because they are rare these likely represent mutants that do not give rise to persistent lineages.

## Changes to spatial-social structure and emergent superspreading

Following pathogen introduction, the mixture of individual-level movement strategies elicits a change in the emergent spatial and social structure at the population level. Pre-introduction populations are spatially clustered near food item patches (*Figure 4A*) due to movement strategies that favour grouping with successful foragers. Pathogen-risk-adapted populations are more dispersed over the landscape, with many individuals found far from food item clusters (*Figure 4B*). This reflects the increased prevalence of the agent-avoiding strategy which leads to a sort of dynamic social distancing. The change in the mixture of population social movement strategies is reflected in the left-skewed degree distributions of pathogen-risk-adapted populations compared to pathogen-risk-naive ones (*Figure 4C*).

We examined the distribution of individual reproductive numbers ($\nu$) from two separate intervals in the simulation: just after pathogen introductions begin ($3000 \leq G \leq 3100$), and 500 generations after introductions begin ($3500 \leq G \leq 3600$). Individual reproductive number distributions from both intervals are strongly left-skewed but have long right-hand tails (up to 12 just after introductions begin; *Figure 4D*). While most infected individuals do not infect any of their neighbours, a small number of these are responsible for a disproportionately large number of further infections, even after the population has adapted to moving under the risk of transmission (*Figure 4D*); this is consistent with the phenomenological definition of superspreading (*Lloyd-Smith et al., 2005*). Our model thus shows how, even in a population with identical individuals that differ only in their movement decision-making

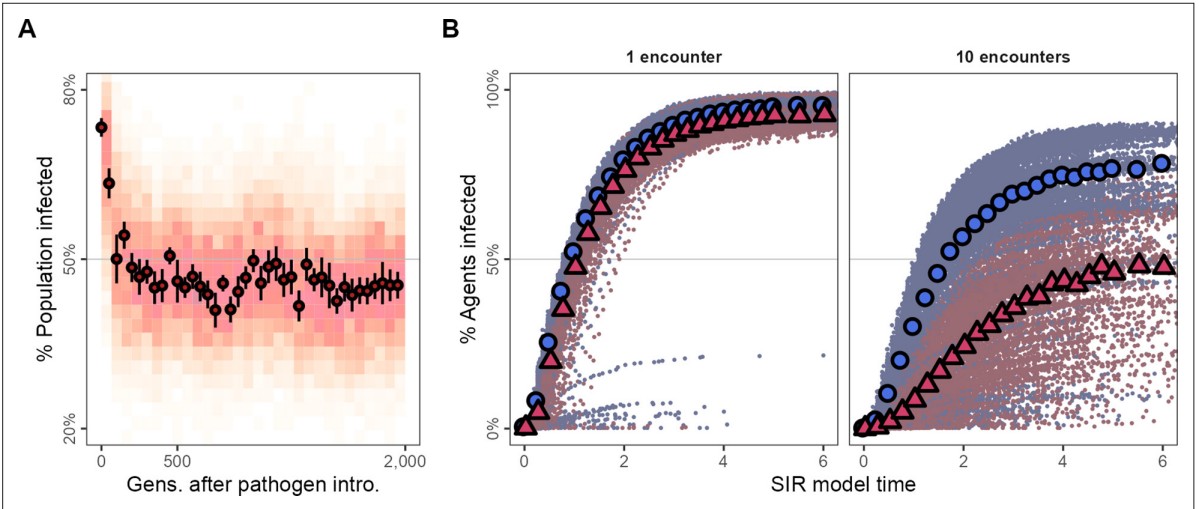

**Figure 5.** Adapting to moving under the risk of pathogen transmission makes populations more resilient to the spread of some kinds of infections. (**A**) In the first generations following pathogen introduction, about 75% the population is infected. However, within 100 generations, tracking the evolutionary shift towards movement strategies that avoid all other individuals, only about 50% of individuals are infected overall. (**B**) The progression of two hypothetical infections, requiring a single encounter, or 10 encounters for a potential transmission, on the emergent social networks of pre- and post-introduction populations. The transmission of the multiple-encounter infection is reduced in populations with disease-adapted movement strategies (pre-introduction: $G$ = 3,000,, blue circles; post-introduction: $G$ = 35000, red triangles). Subfigures in panel (**B**) show means of 25 SIR model replicates (transmission rate $\beta$ = 5.0, recovery rate $\gamma$ = 1.0), run on emergent social network; both panels represent 10 simulation replicates of the default implementation of scenario 1 ($R$ = 2, $\delta E$ = 0.25).

rules, there can be substantial variation in individuals' contribution to the spread of an infectious pathogen.

## Pathogen-risk-adapted movement strategies and the spread of infection

A large majority of individuals in the generations just after pathogen introduction are infected (≈75%; *Figure 5A*). However, tracking the evolutionary change in movement strategies, the number of infected individuals falls to below 50% within 100 generations (*Figure 5A*), remaining low for the rest of the simulation. To examine potential pathogen spread in pre-introduction populations, we ran a simple epidemiological model on the social networks emerging from individuals' movements before and after pathogen introduction (pre-introduction: $G = 500$; post-introduction: $G = 700$). We modelled two infections, (1) first, an infection requiring one encounter, and (2) second, an infection requiring 10 encounters between individuals for a potential transmission event (transmission rate $\beta = 5.0$, recovery rate $\gamma = 1.0$).

Both the single-encounter and multiple-encounter diseases would infect >75% of individuals overall when spreading through the networks of pre-introduction populations (*Figure 5B*). Pathogen-risk-adapted populations' social networks are however more resilient to the multiple-encounter

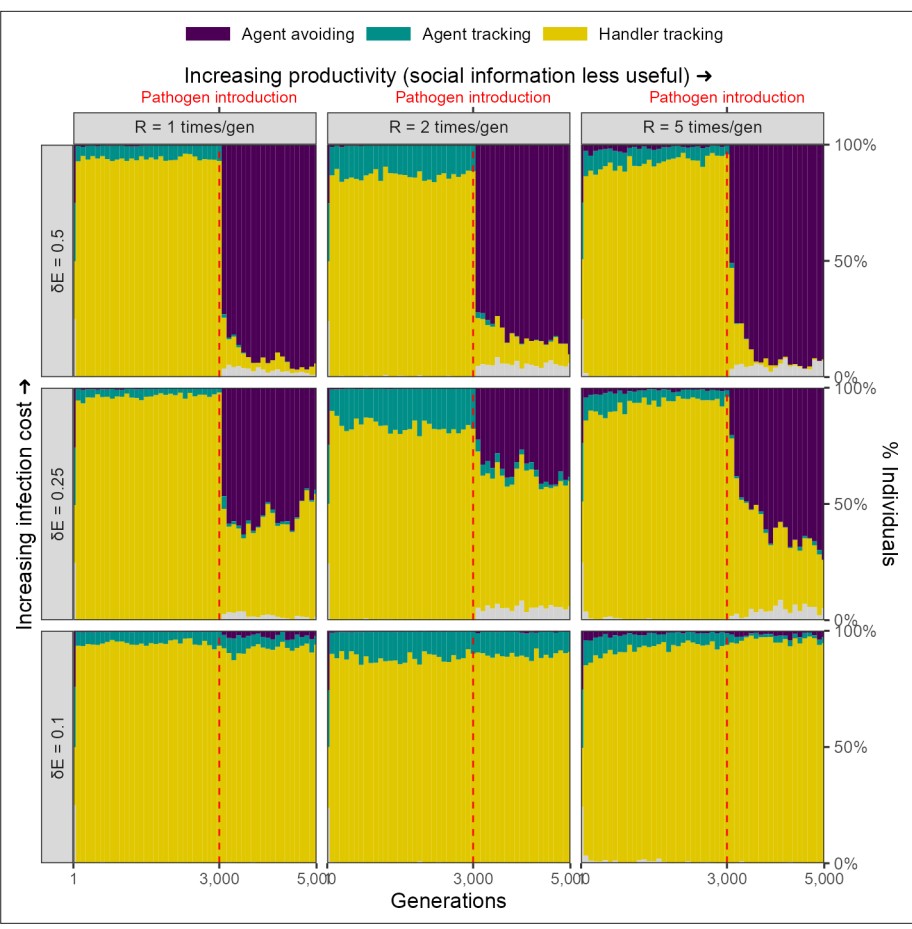

**Figure 6.** Infection cost, but not the usefulness of social information, shapes the rapid evolutionary change in movement strategies triggered by pathogen introduction. Pre-introduction ($G = 3000$; dashed line) populations mostly contain individuals that track successful foragers (handler-tracking), with a small number of individuals that track all foragers (agent-tracking). After pathogen introduction, indiscriminate agent avoidance becomes a common strategy, but only when landscape productivity cannot compensate for infection costs ($\delta E \in 0.25, 0.5$). In cases where the infection cost is low, handler-tracking persists as the most common strategy after pathogen introduction. All panels show frequencies over 10 replicate simulations in 100-generation bins; frequencies are stacked. Grey areas show the relatively uncommon non-handler-tracking strategy that sometimes arises due to mutations.

infection compared to their pre-introduction, pathogen-risk-naive ancestors as these social networks are sparser and individuals are more weakly connected (*Figure 5B*). While nearly all individuals in post-introduction populations would be finally infected by the single-encounter infection the same as their pre-introduction, pathogen-risk-naive ancestors, the spread of the multiple-encounter infection would be substantially reduced in comparison (ever infected: ≈ 50%).

## Effect of landscape productivity and infection cost

For our scenario 1, we further explored the effect of two ecological parameters, landscape productivity ($R \in 1, 2, 5$) and infection cost per timestep ($\delta E \in 0.1, 0.25, 0.5$), on simulation outcomes. Before pathogen introduction, the same social movement strategies evolve on landscapes of all productivity levels (*Figure 6*).

### Infection cost

The introduction of the infectious pathogen leads to a rapid evolutionary shift in social movement strategies, but only in those scenarios in which the cost of infection is substantial ($\delta E \in 0.25, 0.5$). When the cost of infection is low ($\delta E = 0.1$), the handler-tracking strategy persists as the most common social movement strategy. This is because the low infection costs can be compensated by individual intake. In scenarios where infection costs are higher, populations shift away from handler-tracking towards agent avoidance as the former strategy is associated with higher infection risk, and as infection costs are not as easily offset by intake. The frequency of agent avoidance increases with infection cost; while

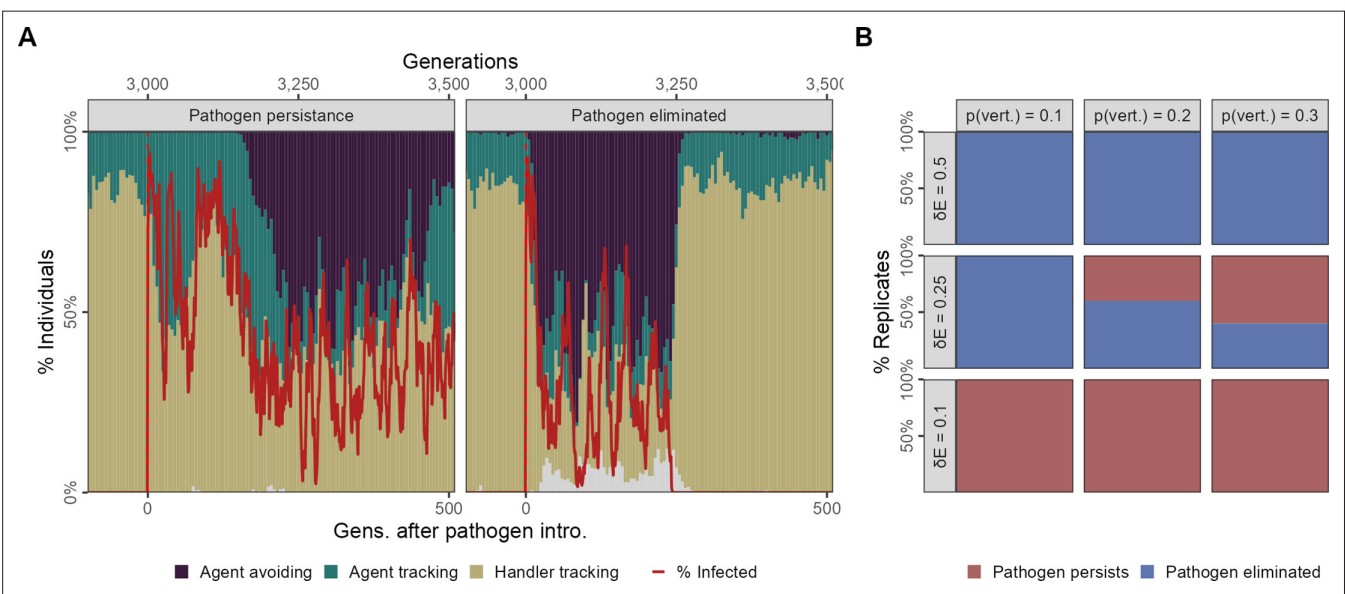

**Figure 7.** Feedback between evolutionary transitions in social movement strategies and pathogen persistence after a single introduction event with vertical transmission. In scenario 2 with only a single introduction event (initial infections = 20) but also vertical transmission from parents to offspring at the reproduction stage, simulation replicates show divergent outcomes. (**A**) In some replicates, the population is slow to transition away from sociality, and the agent-avoiding strategy becomes common only after 200 generations. In such cases, the pathogen persists among social individuals for over 500 generations (panel *Pathogen persistence*). In contrast, when the population undergoes a rapid evolutionary shift and agent avoidance becomes common within 100 generations, the number of infections falls rapidly. This sets up a feedback between social strategies and the number of infections, with infections tracking the frequency of the more social strategies with a time lag of a few generations (panel *Pathogen eliminated*). In some cases, infections drop to zero, which drives the pathogen extinct following which there is an extremely rapid recovery in the frequency of the more social handler-tracking strategy, and the near-complete extinction of agent-avoiding foragers. (**B**) Infection cost and the probability of vertical transmission together influence whether populations undergo evolutionary transitions that lead to pathogen elimination. In general, pathogen elimination is more common when pathogen costs are higher (as infected individuals have fewer offspring), and when the probability of vertical transmission is low. When infection costs are low ($\delta E = 0.1$), there is no evolutionary transition, and the pathogen persists in the population even when transmission between generations is low ($p_v = 0.1$). At intermediate infection costs ($\delta E = 0.25$), pathogen persistence increases with the probability of vertical transmission. All panels show the outcomes of 10 replicates with the default landscape spatial structure, and with a landscape productivity $R$ = 2. Pathogen persistence or elimination is measured at $G = 3500$, that is,, 500 generations after the first introduction.

approximately 40% of all individuals in our default cost case ($\delta E$ = 0.25) are agent-avoiding, nearly all individuals avoid all other foragers when the per-timestep infection cost is doubled ($\delta E$ = 0.5).

## Landscape productivity

The productivity of the resource landscape should be expected to control the usefulness of social information, with social information less useful on more productive landscapes (due to the increased availability of direct cues). We expected that this would lead to greater handler-tracking persisting on lower productivity landscapes, but did not find this to be case; indeed, there did not appear to be an effect of productivity on the evolution of social movement strategies (*Figure 6*).

## Pathogen persistence after a single introduction with vertical transmission

In our scenario 2, we introduced the pathogen only once to 4% (N = 20) individuals in generation 500, and this more closely simulates the sort of introduction that would be expected in a novel, cross-species spillover. Focusing on our default parameter combination ($R$ = 2, $\delta E$ = 0.25, $p_v$ = 0.2), we observed that prior to pathogen introduction the population followed the same ecological and evolutionary principles we laid out for scenario 1, and all replicates were similar (*Figure 7A*). The pathogen is successfully transmitted from parents to offspring in the initial generations following the introduction event and among individuals of the same generational cohort. This produces ecological patterns very similar to scenario 1, with large numbers of infections (*Figure 7A*).

## Evolutionary change can lead to pathogen extinction

However, we observed that replicates begin to differ at this stage in whether the evolutionary change in sociality seen therein is sufficient to drive the pathogen extinct (by reducing its transmission opportunities until no individuals are infected). In some replicates the emergence of agent avoidance is slow, and the frequency of this strategy seldom crosses 50% (*Figure 7A* panel: *Pathogen persistence*). Importantly, this means that the pathogen persists for over 500 generations after the initial introduction, with chaotic dynamics in the number of infections in each generation, which only roughly track changes in the frequency of the agent-avoiding strategy.

In contrast, in some replicates agent avoidance rapidly reaches a prevalence of over half of all individuals. This evolutionary transition away from sociality leads to an initial, corresponding decline in the number of infections in each generation, as expected (*Figure 7A*). The number of infections is reduced to zero within 250 generations, and the pathogen is driven extinct extinction (*Figure 7A* panel: *Pathogen eliminated*). The complete elimination of the pathogen is then associated with an even more rapid recovery of the more social movement strategies prevalent before pathogen introduction handler-tracking and agent-tracking and a near extinction of the agent-avoiding strategy.

## Infection cost and vertical transmission probability influence pathogen persistence

We examined the effect of the per-timestep infection cost ($\delta E$) and the probability of vertical transmission ($p_v$) on whether the pathogen persisted for at least 500 generations through vertical transmission alone, that is, without repeated external introduction events such as in scenario 1. When infection costs are low, there is no evolutionary transition in social movement strategies, and this leads to pathogen persistence in all replicates ($\delta E$ = 0.1; *Figure 7B*). When infection costs are high ($\delta E$ = 0.5), the pathogen is always eliminated within 500 generations (frequently, within 200 generations), with the pathogen persisting longer as $p_v$ increases. This is accompanied by sharp evolutionary transitions towards agent avoidance, which are reversed once the pathogen goes extinct. At intermediate infection costs ($\delta E$ = 0.25), a mixture of outcomes is obtained (*Figure 7B*). When the probability of vertical transmission is low ($p_v$ = 0.1), there is no evolutionary shift in social movement strategies, but the pathogen is eliminated within 250 generations, and the number of generations required for pathogen elimination varies widely. As $p_v$ increases (0.2: default, 0.3), the pathogen persists in more scenario replicates. These results suggest how novel pathogen introductions could lead to pathogens becoming endemic among animal populations.

## Effect of modelling choices on simulation outcomes

Modelling choices can have a substantial effect on the outcomes of simulations with multiple, complex interactions among components (*Scherer et al., 2020*; *Netz et al., 2022*; *Gupte et al., 2023*). We show the effect of varying implementation on some key aspects of our model, with a focus on our scenario 1 (with repeated pathogen introduction): (1) how the infectious pathogen imposes fitness costs, (2) where individuals are initialised, or born, on the landscape relative to their parents' positions (which may be thought of as natal dispersal), (3) whether individuals are allowed to reproduce when they have a negative energy balance, (4) the duration for which social information is available, in the form of changes to the handling time, (5) changes to the spatial structure of the landscape, and (6) the sporadic introduction of the pathogen.

## Infection cost as a reduction in foraging efficiency

We considered an alternative implementation in which the infectious pathogen is considered to reduce an animal's ability to process intake, or to require a portion of daily intake to resist, such that an individual with lifetime intake $N$, has a net energetic gain $E = N \times (1 - \delta E)^t$ after being infected by a pathogen for $t$ timesteps. In this implementation, there is a rapid evolutionary shift in movement strategies after pathogen introduction, similar to that in our default implementation, but only when the costs of infection are relatively high ($\delta E = 7.5\%$), and the usefulness of social information is limited by

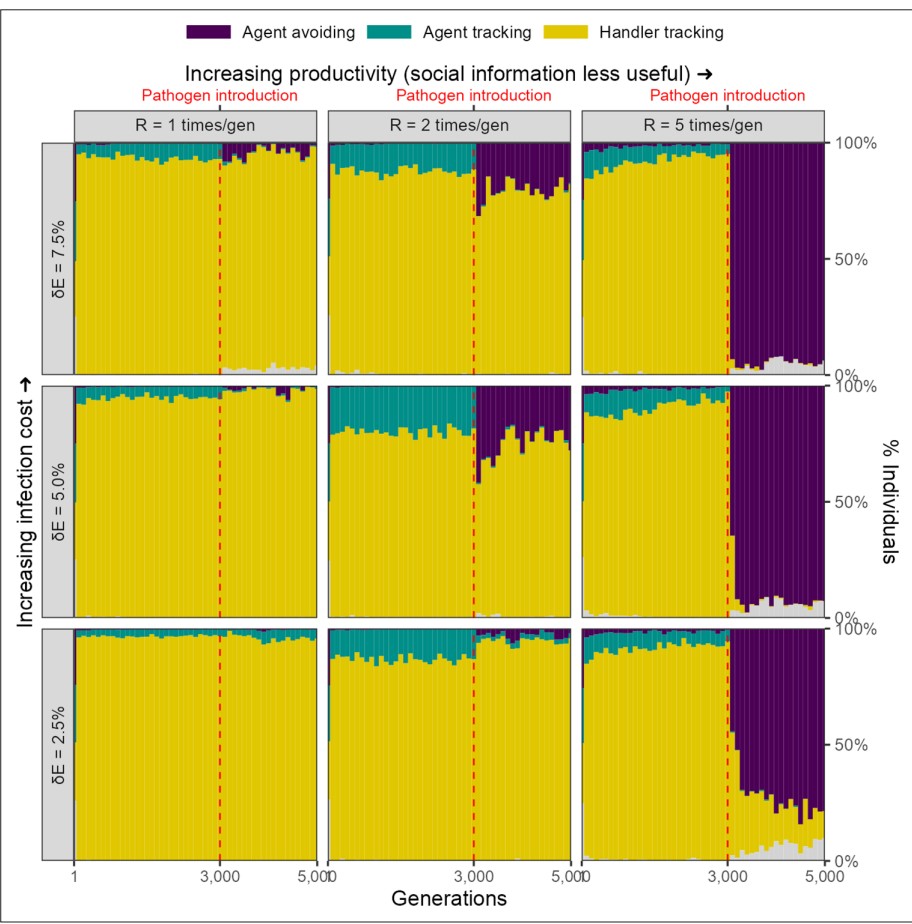

**Figure 8.** Rapid evolutionary change under some conditions in an alternative implementation of disease costs. In our alternative, percentage costs implementation of the infectious pathogen, there is a rapid shift in the mix of movement strategies after pathogen introduction, but only when the costs of infection are relatively high (7.5%), and the usefulness of social information is limited by the abundance of food items ($R = 5$). In these cases, the agent-avoiding strategy is the most common social movement strategy, forming a smaller proportion of the population mixture of social movement strategies when the infection cost is lower, or when the usefulness of social information is greater (lower $\delta E$ and lower $R$, respectively).

the abundance of food items ($R$ = 5). Under these conditions, the agent-avoiding strategy becomes the most common strategy. Under conditions of median landscape productivity and intermediate to high pathogen costs ($\delta E \in$ 5.0% and 7.5%, $R$ = 2), the agent-avoiding strategy also emerges, but forms only a low proportion of the population. Under all other conditions, the handler-tracking strategy continues as the most common strategy (*Figure 8*).

### Large-scale natal dispersal of individuals

Our model implements small-scale or local natal dispersal and individuals are initialised close to their parents' last position a defensible choice as many organisms do not disperse very far from their ancestors. An alternative implementation is to initialise individuals in each new generation at random locations on the landscape (see e.g.; *Gupte et al., 2023*); this can be called large-scale or global natal dispersal. This may be a reasonable choice when modelling animals during a specific stage of their life cycle, such as after arriving on a wintering or breeding site after migration. When animals do not disperse very far, they may adapt their movement strategies to the local conditions which they inherit from their parents (ecological inheritance) (*Badyaev and Uller, 2009*). By forcing animals in each new generation to encounter ecological circumstances potentially different from those of their parents, implementing global dispersal can help investigate whether animals' evolved movement strategies are truly optimal at the global scale (*Gupte et al., 2023*). We implemented global dispersal by running 10 replicates of each parameter combination (three combinations of $\delta E$ = 0.25 and $R \in$ 1, 2, 5; 30 simulations in all), with dispersal set to 10. This means that individuals' initial positions are drawn from a normal distribution with standard deviation = 10, centred on the location of their parent.

We found that our model is broadly robust to implementing large-scale natal dispersal, with the evolutionary outcomes very similar to those seen in our default implementation with small-scale natal dispersal (*Figure 9A*). Most individuals are handler-tracking before the introduction of the novel pathogen, which makes them to gain the benefits of social information on the location of a resource patch (of which handlers are an indirect cue), while avoiding potential competitors, as well as potentially moving away from areas without many food items. After pathogen introduction, there is a rapid evolutionary shift in social movement strategies, with an increase in agent avoidance, similar to the shift seen in our default implementation of local dispersal. The effect of landscape productivity on the mix of proportions of the pre- and post-pathogen introduction strategies does not appear to be significant.

### Energy threshold on reproduction

Individuals may skip reproduction when their body condition is below some threshold, as would be expected when infected by a transmissible pathogen. Restricting reproduction to only those individuals which had a positive energy balance ($\sum \text{intake} > \delta E \sum \text{time infected}$), we found that for our default parameter combination the handler-tracking strategy persists as the most common strategy after pathogen introduction, with agent avoidance making up a small proportion of the population (*Figure 9B*). This is likely because agent-avoiding foragers also avoid food clusters, and thus have low or no intake, which precludes then from reproducing and proliferating. At a lower infection cost ($\delta E$ = 0.1), there is broadly no effect of pathogen introduction on the evolved social movement strategy, and handler-tracking persists at a high frequency. When infection costs are higher ($\delta E$ = 0.5), handler-tracking still persists after pathogen introduction, but with frequent and strong irruptions of agent-avoiding individuals over the generations following introduction.

### Persistence of social information in the form of handling time duration

In our model, the availability of inadvertent social information on the location of food item clusters is controlled by the handling time parameter $T_H$ (default = 5 timesteps). Running our default implementation of scenario 1 ($\delta E$=0.25, $R$ = 2) with four alternative values of handling time 0, 1, 2, and 10, we found that at low handling times ($T_H \in$ 1, 2), the handler-tracking strategy persists as the dominant strategy after pathogen introduction, with a small proportion of agent-avoiding individuals (*Figure 9C*). Doubling handling time ($T_H$ = 10) leads agent avoidance to rapidly become the dominant strategy, likely because the cumulative risk of pathogen transmission from nearby infected individuals increases with increased handling time. These results suggest how the evolution of social information usage can be strongly influenced by its indirect costs (here, transmission risk), although we do

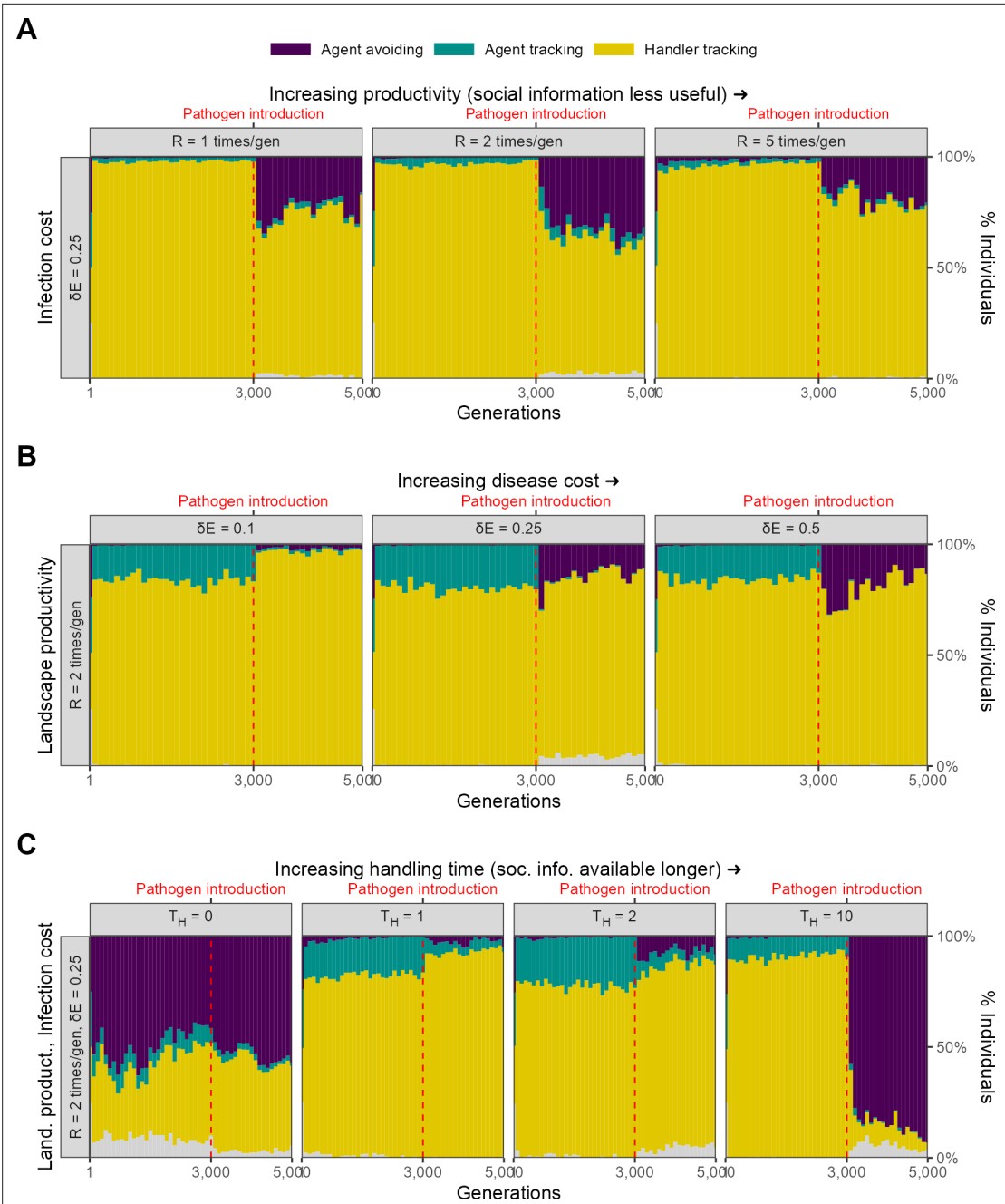

**Figure 9.** Evolutionary outcomes show the effect of modelling choices in alternative implementations of scenario 1. (**A**) Large-scale (or global) natal dispersal leads to evolutionary outcomes similar to the default implementation of small-scale or local natal dispersal ($R \in 1, 2, 5$; $\delta E = 0.25$). (**B**) A threshold on reproduction such that only individuals with a net positive energy balance (lifetime intake > total infection cost) are allowed to reproduce leads to the persistence of the handler-tracking strategy. This is likely because the intake-infection risk trade-off of complete agent avoidance leads to an indirect avoidance of food items, and hence intake; in turn this likely prevents agent-avoiding individuals from reproducing. (**C**) The availability and indirect costs of using social cues jointly determine how the persistence of inadvertent social information affects the evolution of social movement strategies. When the indirect costs of social information are low ($T_H \in 1, 2$), handler-tracking persists beyond pathogen introduction. When these costs increase, individuals eschew social associations and are agent-avoiding ($T_H = 10$). When there is no social information on food items available ($T_H = 0$), all individuals are functionally agent-avoiding (as there are no handlers).

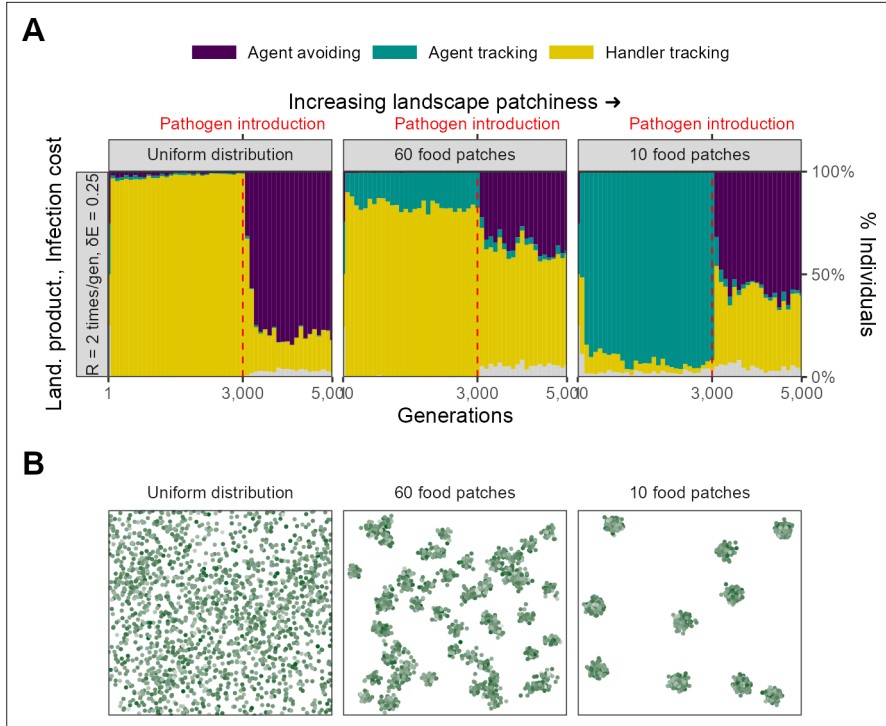

**Figure 10.** Landscape spatial structure influences the evolution of social movement strategies before, but not after, pathogen introduction. (**A**) In two implementations with different spatial structures ($R$=2, $\delta E$ = 0.25), pre-pathogen dynamics are actually more different than post-introduction ones. On landscapes with a uniform food distribution (left panel: *Uniform distribution*), all individuals before pathogen introduction were handler-tracking. On more clustered landscapes (right panel: *10 food patches*; default = 60, centre panel), the rare agent-tracking strategy is most common before pathogen introduction. This is likely because the time cost of moving between distant patches on clustered landscapes is higher than that of exploitation competition. After pathogen introduction, agent avoidance rapidly becomes a common strategy. It is more dominant on uniform landscapes (~80%) likely because the usefulness of social information is lower there. (**B**) Panels show representative landscapes corresponding to the outcomes in (**A**).

recognise that this linkage between social information use and infection risk is particularly strong in our model due to the immobilisation of handling individuals. A more thorough investigation of this link would ideally use a model in which social information can be gained even in the absence of individuals themselves. When there is no handling time ($T_H = 0$), a mixture of handler-tracking and agent-avoiding strategies persists in the population from the beginning of the simulation, with no change following pathogen introduction (*Figure 9C*). In this case, there are never any handlers, and thus oscillations in social movement strategy most likely represent neutral variation around the handler preference $s_H$; most individuals would more accurately be described as non-handler avoiding.

## Spatial structure of the resource landscape

Since ours is a spatial model, and the explicit consideration of space and movement is key to its outcomes, we very briefly examined the effect of landscape spatial structure on the evolutionary outcomes of our scenario 1 (*Figure 10*). We considered two alternative food item distributions: (1) food items distributed uniformly across the landscape, and (2) food items more patchily distributed than the default, with only 10 food item clusters (default = 60). We compared the outcomes on these landscapes with those from our default scenario, with all parameters expect spatial structure kept the same ($R$=2, $\delta E$ = 0.25, $N$ food items = 1800; *Figure 10B*).

Landscape spatial structure influences the mixture of social movement strategies evolved before pathogen introductions (*Figure 10A*). On the uniform landscape, handler-tracking was the most common strategy before pathogen introduction, with nearly all individuals of this strategy. In contrast, on the more patchy landscape, the indiscriminately social agent-tracking strategy was the most

common before pathogen introductions. Both of these are in contrast with our default scenario, in which most individuals were handler-tracking, but with a substantial proportion of agent-tracking individuals.

This overall pattern is likely due to the increasing benefit of social information and the increasing costs of movement between profitable areas of the landscape. As landscapes become more clustered, direct food item cues become more difficult to find, as food items are found in smaller and denser patches. This increases the value of sociality as individuals are likely to found near food item clusters. Furthermore, the indirect costs of movement also increase on patchy resource landscapes as individuals have to pay an increased cost in time (which could have been spent foraging) in moving between food item clusters. In an implementation not formally shown here, the same effect can be achieved by adding a small cost to each movement step; this leads to the evolution of indiscriminate sociality in the form of agent-tracking on the default landscape as well. Overall, both the increasing local density of food items and the costs of movement lead to an increase in agent-tracking as individuals prefer to trade movement costs for the costs of increased local competition for food items.

Following pathogen introduction, populations on both landscapes undergo a rapid evolutionary transition to a mixture of handler-tracking and agent-avoiding strategies, which is similar to the change observed in our default scenario (*Figure 10A*). However, the landscapes differ in the proportions of the two strategies, with agent avoidance more common on the uniform landscape than on the patchy landscape. Interestingly, both of these extremes of landscape structure have more agent-avoiding individuals than our default landscape of 60 food item clusters. On the uniform landscape, this is likely because food items are readily found with the need for indirect social cues, and so most individuals avoid each other. It is less clear why this is the case on the more patchy landscape; it is possible that the denser food item patches lead to more associations and more rapid pathogen spread, with handler-tracking individuals infected for longer periods than agent-avoiding ones, leading to a stronger intake-infection trade-off. Overall, this scenario demonstrates how spatial structure can play an important role in the evolution of social movement strategies, but also how the risk of infection can lead to landscapes with very different spatial structures eventually populated by similar social movement strategies.

## Sporadic introduction of infectious pathogens

Finally, we implemented a variant of our main model, in which the infectious pathogen is introduced only sporadically after the first introduction event (at $G = 3000$). Specifically, we modelled probabilistic introduction of the pathogen in each generation following the initial introduction. We call the per-generation probability of a novel pathogen introduction event the spillover rate, and we ran this model variant for three values of the spillover rate: 0.05, 0.1, and 0.25. Instead of examining the joint effect of landscape productivity and cost of infection as well, we only examined the effect of infection cost, implementing three different variants with an infection cost $\delta E$ of 0.1, 0.25, and 0.5. We kept all other model parameters similar to the default scenario of our main model, and importantly, considered only a landscape productivity $R$ of 2.

Following pathogen introduction, we found that there was little to no change in the population-level mixture of movement strategies in this model variant (*Figure 11*). This is regardless of the probability of a novel pathogen introduction, and the cost of infection by a pathogen. Across the simulation, the most common social movement strategy remains handler-tracking, that is, preferring locations with multiple individuals regardless of their foraging status. Since there is little to no change in social movement strategies, we did not expect nor find changes in ecological outcomes.

## Discussion

Our general model captures important features of infectious pathogen or parasite transmission among host animals in a foraging context that is relevant to many species. Adding an explicit spatial setting has allowed us to more finely probe the effects of individual behavioural variation, pathogen characteristics, and landscape properties on the emergence of animal sociality and the spread of disease. The mechanistic combination of ecological, evolutionary, and epidemiological dynamics in a spatial setting is unprecedented for host movement-disease models (*White et al., 2018b*; *Manlove et al., 2022*). The key feature of our approach is to let the ecological outcomes (intake, time infected)

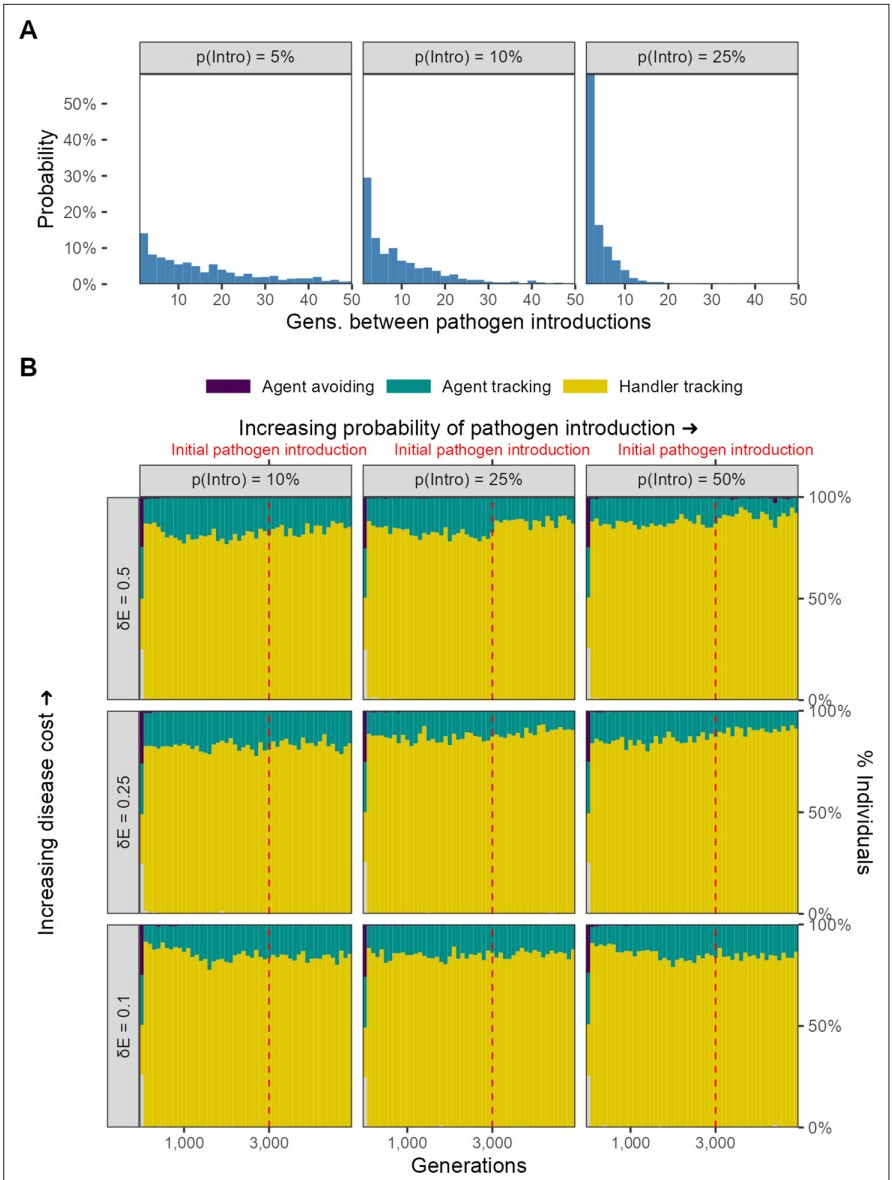

**Figure 11.** No evolutionary change in social movement strategies when novel pathogen introduction events are relatively uncommon. (**A**) In our alternative implementation of scenario 1, the pathogen is only introduced sporadically after the initial introduction ($G$ = 3000; red line in panel **B**). (**B**) When introductions are relatively rare and sporadic, there is no shift in the mixture of movement strategies after pathogen introduction. The handler-tracking strategy remains common across parameter combinations. Panels represent combinations of the per-timestep cost of infection $\delta E$ and the spillover rate (rows), which is the probability of pathogen introduction in each generation (columns). All panels show the combined outcomes of 10 replicate simulations.

of individual social movement decisions in one generation affect the mixture of social movement strategies of the next generation. Our approach shows how host evolutionary dynamics can be incorporated into mechanistic movement-disease models (*Manlove et al., 2022*) and how this approach extends current understanding of the evolutionary causes and consequences of animal spatial and social behaviours (*Albery et al., 2021a*; *Webber and Vander Wal, 2018*; *Webber et al., 2023*; *Romano et al., 2020*; *Romano et al., 2022*; *Kurvers et al., 2014*). To aid in the uptake of our modelling approach, we provide both a written description of the model (see 'Methods') as well as the full, documented source code (see 'Data availability').

Presently, most movement-disease models are non-evolutionary (*White et al., 2018a*; *White et al., 2017*; *Scherer et al., 2020*; *Lunn et al., 2021*; *Manlove et al., 2022*), presumably because evolution

is expected to be too slow to impact epidemiological–ecological outcomes. We demonstrate the pitfalls of this assumption: evolutionary transitions in sociality occur within only a few generations, comparable to the time required for the development of key social aspects of animal ecology, such as migration routes (*Jesmer et al., 2018*; *Cantor et al., 2021*). We also demonstrate the tension inherent to sociality under the risk of an infectious pathogen in an explicitly spatial context. We show how populations, initially evolved to find patchily distributed food using social information, rapidly evolve to become more sensitive to potential infection risk and eschew social encounters when an infectious pathogen is introduced. Our results suggest how qualitatively and quantitatively different social movement strategies making different trade-offs between social information and infection risk can coexist in a single population (*Wolf and Weissing, 2012*; *Webber and Vander Wal, 2018*; *Gartland et al., 2022*; *Webber et al., 2023*; *Wolf et al., 2008*). Furthermore, our model shows how these trade-offs are outcomes of movement decisions, an aspect which would be difficult to study in a non-spatial model.

Following pathogen introduction, the evolutionary shift in social movement strategies is much more rapid than the timescales usually associated with the evolution of complex traits such as sociality (about 100 generations). Avoiding potentially infectious individuals is a key component of navigating the landscape of disgust (*Weinstein et al., 2018*). Our results show that sensitivity to cues of high pathogen transmission risk can rapidly evolve following the introduction of a novel pathogen. The emergence of qualitative individual variation in social movement strategies, and especially the trade-off between movement, associations, and infection risk, also demonstrates the evolution of sociability as a personality trait (*Gartland et al., 2022*). We also find substantial individual variation in the quantitative importance of social cues overall, which is a key component of the evolution of large-scale collective behaviours, such as migration (*Guttal and Couzin, 2010*). Our work suggests how, by leading to the necessary diversity in social movement strategies, a novel pathogen may actually lay the groundwork for the evolution of more complex collective behaviour. Nonetheless, the rapid decreases in social interactions should primarily prompt concern that the evolutionary consequences of pathogen introduction could slow the transmission of, and erode, animal culture (*Cantor et al., 2021*) including foraging (*Klump et al., 2021*) and migration behaviours (*Jesmer et al., 2018*; *Guttal and Couzin, 2010*).

Pathogens themselves typically have shorter generation times than their hosts and may also evolve rapidly in response to changes in host sociality (*Prado et al., 2009*; *Ashby and Farine, 2022*; *Bonds et al., 2005*). Our aim was to investigate how host behaviour evolved according to a predetermined (but varied) suite of pathogen characteristics across different simulation runs. Furthermore, we wanted to examine the effects of *introduction events* which are expected to become more common (*Carlson et al., 2022a*), but which need not necessarily lead to the pathogen becoming endemic in a population. Holding the pathogen traits steady and unable to evolve in the course of a simulation is thus a necessary choice in order to gain these first tangible insights from our model. Allowing simultaneous antagonistic coevolution between trophic levels, such as hosts and pathogens or predators and prey, could exponentially complicate the findings of a given eco-evolutionary model, such as by producing generationally staggered outcomes or cyclical Red Queen patterns (*Prado et al., 2009*; *Netz et al., 2022*), and can require much longer runs to attain stationary results or to identify optimal strategies. However, pathogen evolution in response to host behaviour is something that we would be excited to investigate in the future using this modelling framework. Indeed, a mixture of host social strategies could allow for the maintenance of a corresponding diversity in pathogen strategies as well (*Ashby and Farine, 2022*; *Prado et al., 2009*) as is also seen in predator–prey co-evolution (*Netz et al., 2022*). One conceptual impediment is modelling pathogen traits in a mechanistic way. For example, it is widely held that there is a trade-off between infection cost and transmissibility with a quadratic relationship between them (*Ashby and Farine, 2022*; *Bonds et al., 2005*; *Prado et al., 2009*), but this is a pattern reported from empirical studies and not a process per se. A tractable starting point might be to adapt our scenario 2 with vertical transmission to examine the evolution of pathogen traits that influence both transmissibility and virulence with an unchanging host (such as an adaptation of *Lion and Boots, 2010*).

In our model, landscape productivity ($R$) is a proxy for the usefulness of sociality overall as social information is less useful when direct resource cues are abundant (high $R$; see also) (*Gupte et al., 2023*). Social information benefits in disease models often have no mechanistic relationship with the

subject of the information (e.g. food or predators; *Ashby and Farine, 2022*). In contrast, social information benefits in our model are emergent outcomes of animal movement and foraging behaviour which is only possible due to the explicit spatial nature of our model. It is surprising then that landscape productivity does not strongly influence the evolution of social movement strategies, but this may yet be an important factor in enabling high-movement, low-infection strategies when movement is inherently costly. In our model, movement has an indirect time cost; moving away from food items leaves less time in which to make up fitness differences with other individuals through foraging. This is essentially why we find that landscape spatial structure strongly influences the mixture of social strategies evolved before pathogen introduction. However, we found that across a spectrum of spatial structures, pathogen introduction resulted in a convergence in social movement strategies; this evolutionary component may be an important consideration in studies of how spatial structure can influence the spread of infection (*He et al., 2021*; *Scherer et al., 2020*; *White et al., 2018c*; *White et al., 2017*). Furthermore, movement can be an energetically demanding process that could influence whether dynamic social distancing to avoid infection risk, as evolved in our model, would be a viable movement strategy. Future extensions of our model could add a small cost to movement in order to explore the interplay of landscape productivity and spatial structure in determining direct indirect movement costs and the consequences for social movement strategies.

Infection costs do affect which social movement strategies evolve in our model and may help explain intra- and inter-specific diversity in social systems across gradients of infection costs (*Altizer et al., 2003*; *Sah et al., 2018*). Studies tracking social movements and the potential for pathogen spread could form initial tests of our basic predictions (*Wilber et al., 2022*). Our model suggests that animal populations may be able to adapt relatively quickly to the spillover and eventual persistence of infectious pathogens, even when they cannot specifically detect and avoid infected individuals (*Altizer et al., 2003*; *Stroeymeyt et al., 2018*; *Stockmaier et al., 2021*; *Pusceddu et al., 2021*). While the most noticeable effect of pathogen outbreaks is mass mortality (*Fey et al., 2015*), even quite serious pathogens such as sarcoptic mange (*Almberg et al., 2015*), foot-and-mouth disease (*Jolles et al., 2021*; *Bastos et al., 2000*; *Vosloo et al., 2009*), SARS-CoV-2 (*Chandler et al., 2021*; *Kuchipudi et al., 2022*), and avian influenza (*Global Consortium for H5N8 and Related Influenza Viruses, 2016*; *Wille and Barr, 2022*), among others, appear to spread at sublethal levels for many years between lethal outbreaks. Our model shows how population-level behavioural changes could occur even without mortality effects due to evolutionary shifts in sociality alone. The pathogen-risk-adapted population in our model are unable to escape infection entirely and have significantly worse net energy per-capita (just over zero), which could leave them vulnerable to extreme ecological conditions. Our work suggests that decreased sociality resulting from adaptation to a novel pathogen could slow the transmission of future novel pathogens. While decreased sociality could also reduce the prevalence of previously endemic pathogens adapted to a more social host, it may also degrade social immunity through reduced sharing of beneficial commensal microbes, or of low, immunising doses of pathogens (*Ezenwa et al., 2016*; *Almberg et al., 2015*).

The results of our scenario 1 are contingent upon sustained introduction of the pathogen (or its novel strains) to host populations. More sporadic introductions (once every few generations) apparently do not cause evolutionary shifts in social movement. Our scenario 2, which includes transmission from parents to offspring, suggests a mechanism by which such sporadic events, or even a single cross-species spillover event, could have far-reaching evolutionary consequences. Such vertical transmission is believed responsible for the circulation of foot-and-mouth disease in African buffalo (*Jolles et al., 2021*) and of mange among wolves (*Almberg et al., 2015*). Pathogen persistence across a broad swath of parameter combinations for scenario 2 suggests that even single introduction events can lead to a population rapidly becoming a novel source of transmission (loosely speaking, a reservoir) for other, overlapping species. Such dynamics would likely be increased should vertical transmission be coupled with multiple, sporadic pathogen or parasite introductions, which appear to be common in nature (*Levi et al., 2012*; *Jolles et al., 2021*; *Vosloo et al., 2009*; *Bastos et al., 2000*; *Scherer et al., 2020*; *Global Consortium for H5N8 and Related Influenza Viruses, 2016*; *Wille and Barr, 2022*). By demonstrating the multiple ways in which pathogens can affect an animal population, our model suggests how disease is a powerful selective force in favour of detecting and avoiding infection risk cues (*Weinstein et al., 2018*), among which are social cues.

We note that the pathogen characteristics (infection cost) as well as the probability of vertical transmission affect the evolutionary dynamics in scenario 2. In the context of our model, the latter could be interpreted as a factor influencing the association between parents and offspring, such as the length of parental care. This suggests that a directly transmitted novel pathogen should become established readily in species with greater social associations between generations, such as parental care of young *Chakarov et al., 2015*; this may, however, be counteracted by suites of infection-risk-reducing behaviours on the part of adults (*Stroeymeyt et al., 2018*; *Ratz et al., 2021*). Positively, we also find that when the pathogen is eliminated from the population, there is a near instantaneous shift towards (or recovery in) animal sociality. This suggests that if pathogens are extirpated from parts of their former ranges (due to a range of mechanisms, with climatic change as an influence) (*Carlson et al., 2022b*), some animal populations may show a hitherto unexpected increase in sociality, and potentially, novel social behaviours and structures or other aspects of animal culture. Our findings thus suggest an additional consideration when thinking about implementing campaigns that seek to reduce wildlife disease burdens, such as through wildlife immunisation (*Barnett and Civitello, 2020*; *Ezenwa and Jolles, 2015*).

In order to be widely applicable to diverse novel host–pathogen introduction scenarios, our model necessarily makes quite general assumptions. For example, our individuals use both personal and inadvertent social information whenever it is available; even though animals' use of information sources does depend on their behavioural context, this could be examined more thoroughly in future implementations. A wide diversity of pathogens and their dynamics remains to be accurately represented in individual-based models (*White et al., 2017*; *White et al., 2018a*; *Scherer et al., 2020*; *Lunn et al., 2021*). Our framework could be expanded and specifically tailored to real-world situations in which animal populations are exposed to novel pathogens (or strains) that transmit between individuals (*Scherer et al., 2020*; *Wille and Barr, 2022*; *Bastos et al., 2000*; *Jolles et al., 2021*; *Chandler et al., 2021*; *Kuchipudi et al., 2022*). Such detailed implementations could include aspects of the pathogen life cycle (*White et al., 2018b*; *White et al., 2017*), account for sociality as a counter to infection costs (*Ezenwa et al., 2016*; *Almberg et al., 2015*), or model host–pathogen sociality-virulence co-evolution (*Ashby and Farine, 2022*; *Prado et al., 2009*; *Bonds et al., 2005*). Our work could serve as a good base for future models that focus on the importance of other factors, especially more nuanced implementations of reproduction and demography on the evolution of spatial-social strategies under infection risk. For instance, allowing sexual reproduction and considering the effects of infection status on mate choice or limiting pairing to nearby individuals could help explore how individual movement decisions can scale up to speciation and community assembly (*Getz et al., 2016*; *Getz et al., 2015*). Future empirical extensions of our work would ideally combine wildlife monitoring and movement tracking across gradients of pathogen prevalence to detect novel cross-species spillovers (*Chandler et al., 2021*; *Kuchipudi et al., 2022*) and study the spatial and epidemiological consequences of animal movement strategies (*Bastille-Rousseau and Wittemyer, 2019*; *Wilber et al., 2022*; *Monk et al., 2022*). Our model shows why it is important to consider evolutionary responses in movement-disease studies and provides a general framework to further the integration of evolutionary approaches in wildlife spatial epidemiology.

## Methods

We implemented an individual-based simulation model to represent foraging animals (foragers) making movement decisions in an explicit spatial context. Individuals seek out discrete, immobile, depletable food items from which they gain energy that can be devoted to reproduction (similar to capital breeding; see *Figure 1*; *Spiegel et al., 2017*; *Gupte et al., 2023*). Food items are distributed over a two-dimensional, continuous-space resource landscape with wrapped boundaries (a torus). Our model, similar to earlier IBMs with both ecological and evolutionary dynamics (*Getz et al., 2015*; *Netz et al., 2022*; *Gupte et al., 2023*), has two distinct timescales: (1) an ecological timescale comprising of $T$ timesteps that make up one generation ($T$=100 by default), and (2) an evolutionary timescale consisting of 5000 generations ($G$). At the ecological timescale, individuals perceive cues from their local environment: the presence and numbers of food items and other individuals. Individuals make movement decisions according to their inherited movement strategies (see below), and when chancing upon food items, consume them. At the same timescale, individuals that carry an infectious, fitness-reducing pathogen may, when in close proximity with uninfected individuals, pass on the pathogen

with a small probability (see 'Pathogen introduction, transmission, and infection cost'). At the evolutionary timescale, individuals reproduce and transmit their inherited cue preferences, and hence their movement strategies (see 'starting location and inheritance of movement rules') to their offspring. The number of offspring is linked to individuals' success in finding and consuming food items, and to the duration that they were infected by the pathogen at the ecological timescale; this is in line with the replicator equation (*Hofbauer and Sigmund, 1988*). The model was implemented in R and C++ using Rcpp (*R Development Core Team, 2020*; *Eddelbuettel, 2013*) and the *Boost.Geometry* library for spatial computations (https://www.boost.org/); see the 'Data availability' statement for the code archive and development repository.

## Distribution of food items

Our landscape of 60 × 60 units contains 1800 discrete food items, which are clustered into 60 resource patches, for a resource density of 0.5 items per unit area$^2$ (see *Figure 1*). Each available food item can be perceived and harvested by nearby foraging individuals (see below). Once harvested, another food item is regenerated at the same location after a fixed regeneration time $R$, which is set at 50 timesteps by default; alternative values of 20 and 100 represent high and low productivity landscapes, respectively. Food item regeneration is decoupled from population generations, and the actual number of available food items is almost always in flux. In our figures and hereafter, we chose to represent $R$ as the number of times a food item would regenerate within the timesteps in a single generation $T$ (default = 100), resulting in $R$ values of 1, 2, and 5 for regeneration times of 100, 50 (the default), and 20 timesteps. Items that are not harvested remain on the landscape until they are picked up by a forager. Each food item must be processed, or handled, by a forager for $T_H$ timesteps (the handling time, default = 5 timesteps) before it can be consumed (*Ruxton et al., 1992*; *Gupte et al., 2023*). The handling time dynamic is well known from natural systems in which there is a lag between finding and consuming a food item (*Ruxton et al., 1992*).

## Individual foraging and movement

### Foraging

Individuals forage in a randomised order, harvesting an available food item selected at random within their movement and sensory range ($d_S = d_M$, a circle with a radius of 1 unit; see *Figure 1C*). Once harvested, the item is no longer available to other individuals, leading to exploitation competition among nearby foragers. Furthermore, the location of the item also yields no more cues to other foragers that an item will reappear there, reducing direct cues by which foragers can navigate to profitable resource patches. Individuals that harvest a food item must handle it for $T_H$ timesteps (default = 5 timesteps), while all individuals not handling a food item are considered to still be searching for food (*Ruxton et al., 1992*; *Gupte et al., 2023*). While handling, individuals are immobilised at the location where they encountered the item, and thus they may be good indirect indicators of the location of a resource patch (inadvertent social information) (*Danchin et al., 2004*; *Romano et al., 2020*; *Gupte et al., 2023*). Once individuals finish handling a food item, they return to the non-handling, searching state, and are again able to make movement decisions.

### Movement

Our model individuals' movement follows a step-selection framework, wherein the direction of each step is chosen based on the individuals' assessment of local environmental cues (*Fortin et al., 2005*). This assessment is made using inherited movement preferences (as in *Netz et al., 2022*; *Gupte et al., 2023*), which are essentially similar to step-selection coefficients (*Fieberg et al., 2021*). First, individuals scan their current location, and five equally spaced points around their position, at a distance of 1 unit for three cues ($d_S$, see *Figure 1*). These are the number of food items ($F$), the number of foragers handling a food item (handlers: $H$), and the number of idle foragers not handling a food item (non-handlers: $N$). While an individual's count of food items is its personal information, the behavioural status of its neighbours is inadvertent social information; more handlers suggest a large resource patch, while many non-handlers might mean that there is no nearby resource patch. Individuals assign a suitability score to their current position and to each of the five locations using their inherited preferences for each of the cues: $S = s_F F + s_H H + s_N N + \epsilon$ (see also *Netz et al., 2022*; *Gupte et al., 2023*).

The preferences $s_F$, $s_F$, and $s_N$ for each of the three cues are heritable from parents to offspring, while $\epsilon$ is a very small error term drawn for each location, to break ties among locations.

Individual-level combinations of step-selection coefficients estimated from animal tracking data can be used to cluster animals in a behavioural trait space (*Bastille-Rousseau and Wittemyer, 2019*), and we used a similar method to classify our model individuals' movement strategies based on their cue preferences. Since individuals may differ in their inherited preferences for each of the three cues, two individuals at the same location may make quite different movement decisions based on the same local cues. We recognise that real individuals can change their reliance on personal or social information through their lives depending on the behavioural context, but here we chose to focus on the evolutionary timescale, such that the importance of social information was fixed over the lifetime of an individual. All individuals move simultaneously to the location to which they have assigned the highest suitability; this may be their current location, in which case individuals are stationary for that timestep. We modelled individuals as moving in small, discrete steps of fixed size ($d_M$ = 1 unit); this helped us reduce the complexity of the model and to focus on decision-making. Handlers, however, are considered immobile and do not make any movement decisions.

## Pathogen introduction, transmission, and infection cost

Our population evolves for 3/5th of the simulated generations (until $G$ = 3000; of 5000) in the absence of a pathogen, after which a pathogen is first introduced to a randomly selected 4% of individuals (N = 20; primary infections). In scenario 1, the pathogen is then introduced to 20 randomly selected individuals in each generation until the end of the simulation ($G$ = 5000). Novel pathogen introductions can periodically re-occur in natural environments from infected individuals of other spatially overlapping species (e.g. *Kuchipudi et al., 2022*; *Wille and Barr, 2022*; *Chandler et al., 2021*; *Vosloo et al., 2009*; *Bastos et al., 2000*; *Monk et al., 2022*; *Keeling et al., 2001*; *Carlson et al., 2022a*). This is necessary to kick-start the pathogen-movement eco-evolutionary feedback dynamics in each generation as our default scenario has no vertical transmission of the pathogen from parents to offspring. Here, we must emphasise that current knowledge about the frequency of cross-species transmission events in wildlife is extremely poor, yet recent high estimates of SARS-CoV transmission between bats and humans alone (*Sánchez et al., 2022*) make it a plausible assumption that such events are even more common in wildlife. That populations may indeed repeatedly acquire novel pathogens (or strains) from other spatially overlapping species or populations is indeed borne out in a number of studies (e.g. *Vosloo et al., 2009*; *Bastos et al., 2000*; *Monk et al., 2022*; *Keeling et al., 2001*; *Chandler et al., 2021*; *Kuchipudi et al., 2022*), and is especially reinforced by the ongoing outbreak of avian influenza in multiple waterbird species across Eurasia and North America (*Wille and Barr, 2022*).

We sought to capture some essential features of pathogen or parasite transmission among animals (*White et al., 2017*): the pathogen transmits probabilistically from infected host individuals to their susceptible neighbours with a per-timestep probability $p$ = 0.05. This transmission is only possible when the two individuals are within the transmission distance, $d_\beta$. For simplicity, we set $d_\beta$ to be the movement range (1 unit). Once transmitted, the pathogen is assumed to cause a chronic infection which reduces host energy stores by a fixed amount called $\delta E$ in every following timestep; $\delta E$ is set to 0.25 by default (alternative values: 0.1, 0.5). In our default scenario, this means that individuals once infected do not increase their net energetic balance as they lose more energy per timestep to the disease than they can gain from foraging (but note scenarios with lower $\delta E$ where this is not the case). We also considered an alternative implementation of disease costs: instead of imposing an absolute energetic cost that is independent of intake, infection reduces energy gained through intake by a certain percentage, decreasing the value of each food item. This may be thought of as infection-reducing foraging efficiency or as requiring some proportion of intake to be devoted to immune resistance rather than (eventually) being given over to reproduction.

Recognising that novel pathogen spillovers in each generation represent a somewhat extreme scenario, we also considered implementations in which pathogen introductions only occur sporadically in the generations after the initial event, rather than in every generation. Furthermore, in scenario 2 we modelled only a single introduction event, but allowed infected parents to pass the pathogen on to any offspring with a one-time probability $p_v$ = 0.2 (which we refer to as vertical transmission; alternative values: 0.1, 0.3). We deliberately set $p_v > p$ to reflect that offspring in early life may be in close contact with their parents, providing ample opportunity for pathogens to transmit. We would

note that vertical transmission can occur only once as generations change; this is in contrast with (horizontal) transmission between foragers, which has a *per-timestep* probability.

## Starting location and inheritance of movement decision-making rules

We considered a population of haploid individuals with discrete generations that do not overlap with each other in practical terms and which have asexual inheritance to reduce model complexity. At the end of each parental generation, we determined the net lifetime energy of each individual as the difference of the total energy gained through food intake and the energy lost through infection. The parental population produces an offspring population (of the same size) as follows: each offspring is assigned a parent at random by a weighted lottery, with the weights proportional to each parent's lifetime net energy (an algorithm following the replicator equation) (*Hofbauer and Sigmund, 1988*; *Hamblin and Hansen, 2013*). This way, the expected number of offspring produced by a parent is proportional to the parents' lifetime success (*Hofbauer and Sigmund, 1988*). We also considered an alternative implementation (for scenario 1 only) in which only individuals with a positive net energetic balance could reproduce.

The movement decision-making cue preferences $s_F$, $s_H$, and $s_N$ are subject to independent random mutations with a probability of 0.01. The mutational step size (either positive or negative) is drawn from a Cauchy distribution with a scale of 0.01 centred on zero. Thus, while the majority of mutations are small, there can be a small number of very large mutations. As in real ecological systems, individuals in the new generation are initialised around the location of their parent (within a standard deviation of 2.0), and thus successful parents give rise to local clusters of offspring (with an alternative implementation where dispersal had a standard deviation of 10.0 units).

## Model output and analysis

### From cue preferences to social movement strategies

To understand the evolution of movement decision-making, and especially how individuals weighed social information, we recorded the populations' evolved cue preferences in every second generation and interpreted them following the behavioural hypervolume approach (*Bastille-Rousseau and Wittemyer, 2019*). When individuals move by step-selection as in our models, the value of each cue preference $s_x$ for $x \in F, H, N$ *relative* to the other cue preferences is more important than the absolute value of any cue preference by itself. Thus individuals that have relatively similar values of all three cue preferences may be thought of as weighing, or preferring each cue relatively equally (or indeed avoiding, if any $s_x < 0$). The relative values of each individual's cue preferences *taken together* may be thought of as the individual *movement strategy*.

To interpret the evolved movement strategies, we first normalised individuals' cue preferences ($s_x$ for $x \in F, H, N$) within the range ($-1, +1$) by dividing each preference by the sum of the absolute values of each preference: $s_x / (|s_H| + |s_N| + |s_F|)$. For example, normalised values of $s_F \approx +1.0$ would indicate a very strong preference for food items, with locations with many food items getting a higher suitability score than locations with fewer food items. Similarly, normalised values of $s_N \approx -1.0$ would indicate a very strong aversion for non-handlers or foragers who have not yet found food. To understand the evolution of individual preferences for social information the presence and status of competing foragers, we began by classifying individuals into four social movement strategies: (1) agent-avoiding, if $s_H, s_N < 0$; (2) agent-tracking, if both $s_H, s_N > 0$; (3) handler-tracking, if $s_H > 0, s_N < 0$; and (4) non-handler-tracking, if $s_H < 0, s_N > 0$. We calculated the relative importance of social cues overall $H, N$ to each individual's movement strategy as $SI_{imp} = (|s_H| + |s_N|)/(|s_H| + |s_N| + |s_F|)$, with higher values indicating a greater importance of social cues.

### Constructing proximity-based social networks

We sought to understand how changes in the frequencies of individual-level movement strategies would affect the broader social and spatial structure of our population. To do this, we created a proximity-based adjacency matrix by counting the number of times each individual was within the sensory and pathogen transmission distance $d_\beta$ ($= d_S, d_M = 1$ unit) of another individual (*Whitehead, 2008*; *Wilber et al., 2022*). We transformed this matrix into an undirected social network weighted by the number of pairwise spatial associations: in a pairwise encounter, both individuals were considered to have associated with each other (*White et al., 2017*). The strength of the connection between any

pair was the number of times the pair were within $d_\beta$ of each other over their lifetime. We logged associations and constructed social networks after every 10% of the total generations (i.e. every 500th generation), and at the end of the simulation. Constructing these networks also allowed us to examine whether changes in social contact patterns could have any effect on the spread of infection in pathogen-naive populations, as against their pathogen-adapted descendants. We also recorded the source of infection for each individual in each generation in which we collected data. The infection source is the infected individual which passed the pathogen on to the focal individual. We used this data to determine the individual reproductive number $\nu$ in order to examine emergent individual variation in pathogen transmission and the potential presence of superspreading (*Lloyd-Smith et al., 2005*).

### Model analysis

We ran 10 replicates of each parameter combination that we present and included the results from all replicates when interpreting simulation outcomes (see 'Data availability'). For both scenarios 1 and 2, we plotted the mix of social information-based movement strategies evolved across generations in each parameter combination. We focused our analysis on scenario 1 and its default parameter combination ($\delta E$=0.25, $R$ = 2), and visualised the mean per-capita distance moved and mean per-capita encounters with other foragers. We examined how the three main social movement strategies agent avoidance, agent-tracking, and handler-tracking changed in frequency over generations. We also examined differences among strategies in the movement distance, associations with other agents, and frequency of infection. We visualised the proximity based social networks of populations in scenario 1 ($\delta E$ = 0.25, $R$ = 2), focusing on generations before and after the pathogen introduction events begin (pre-introduction: $G$ = 3000; post-introduction: $G$ = 3500). We plotted the final size of the outbreak (the total numbers of individuals infected) in each generation after pathogen introduction to examine whether evolutionary changes in movement strategies actually reduced infection spread. We also ran simple network epidemiological models on the emergent individual networks in generations 3000 and 3500 (*Bailey, 1975*; *White et al., 2017*; *Stroeymeyt et al., 2018*; *Wilber et al., 2022*) for robust comparisons of potential pathogen spread in pathogen-risk-naive and pathogen-risk-adapted populations, respectively.

## Acknowledgements

We thank Jan Kreider for helpful feedback on an early draft of the manuscript, Thijs Janssen for help with the simulation model code, and Damien Farine and three anonymous reviewers whose constructive feedback substantially improved the manuscript. We thank the Center for Information Technology of the University of Groningen for providing access to the Peregrine high-performance computing cluster to run simulations. PRG was supported by an Adaptive Life Programme grant made possible by the Groningen Institute for Evolutionary Life Sciences (GELIFES). JG was supported by a grant from the Netherlands Organization for Scientific Research (NWO-ALW; ALWOP.668). FJW and PRG acknowledge funding from the European Research Council (ERC Advanced Grant No. 789240). GFA was supported by the Wissenschaftskolleg zu Berlin with a College for Life Sciences Fellowship, and by the National Science Foundation, USA (DEB-2211287).

## Additional information

### Funding

| Funder | Grant reference number | Author |
|---|---|---|
| European Research Council | Advanced Grant No. 789240 | Pratik Rajan Gupte Franz J Weissing |
| Nederlandse Organisatie voor Wetenschappelijk Onderzoek | NWO-ALW | Jakob Gismann |

| Funder | Grant reference number | Author |
|---|---|---|
| Nederlandse Organisatie voor Wetenschappelijk Onderzoek | ALWOP.668 | Jakob Gismann |
| Wissenschaftskolleg zu Berlin | College for Life Sciences Fellowship | Gregory F Albery |
| National Science Foundation | DEB-2211287 | Gregory F Albery |

The funders had no role in study design, data collection and interpretation, or the decision to submit the work for publication.

## Author contributions
Pratik Rajan Gupte, Conceptualization, Data curation, Software, Formal analysis, Investigation, Visualization, Methodology, Writing – original draft, Writing – review and editing; Gregory F Albery, Amy Sweeny, Methodology, Writing – review and editing; Jakob Gismann, Conceptualization, Investigation, Methodology, Writing – review and editing; Franz J Weissing, Conceptualization, Resources, Supervision, Funding acquisition, Investigation, Methodology, Writing – review and editing

## Author ORCIDs
Pratik Rajan Gupte ⓘ https://orcid.org/0000-0001-5294-7819
Jakob Gismann ⓘ https://orcid.org/0000-0002-2570-590X

## Decision letter and Author response
Decision letter https://doi.org/10.7554/eLife.81805.sa1
Author response https://doi.org/10.7554/eLife.81805.sa2

# Additional files

## Supplementary files
• MDAR checklist

## Data availability
This manuscript presents the results of a simulation model study, and no real data were generated. The Pathomove simulation model code (v.1.2.0) is available on Zenodo at https://doi.org/10.5281/zenodo.7789072 (*Gupte, 2023*) and on Github at https://github.com/pratikunterwegs/pathomove. Code to run the simulations and analyse the output is on Zenodo at https://doi.org/10.5281/zenodo.7789079 and on Github at: https://github.com/pratikunterwegs/patho-move-evol (v.1.1.0). The data presented in this manuscript are also archived on Zenodo with the DOI https://doi.org/10.5281/zenodo.7789060.

The following datasets were generated:

| Author(s) | Year | Dataset title | Dataset URL | Database and Identifier |
|---|---|---|---|---|
| Gupte PR | 2023 | Reference data from the Pathomove simulation, for the manuscript "Novel pathogen introduction triggers rapid evolution in animal social movement strategies" | https://doi.org/10.5281/zenodo.7789060 | Zenodo, 10.5281/zenodo.7789060 |
| Gupte PR, Albery GF, Gismann J, Sweeny AR, Weissing FJ | 2023 | Source Code and Supplementary Material for "Novel pathogen introduction triggers rapid evolution in animal social movement strategies" | https://doi.org/10.5281/zenodo.7789079 | Zenodo, 10.5281/zenodo.7789079 |

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
