## [Editor Report]

This study provides important new insights into the effects that disease can have on movement strategies in animals. The theoretical model that forms part of this contribution generates useful predictions that are widely applicable. In doing so, it will have a lasting impact on the field.

---

## [Decision Letter]

**Decision letter after peer review:**

Thank you for submitting your article "Novel pathogen introduction triggers rapid evolution in animal social movement strategies" for consideration by *eLife*.

Your article has been reviewed by three peer reviewers, including Damien Farine as Guest Reviewing Editor and Reviewer #1, and Christian Rutz as the Senior Editor.

The reviewers have discussed their reviews with one another, and the Reviewing Editor has drafted this decision letter to help you prepare a revised submission.

Essential revisions:

(1) The reviewers were all concerned about the lack of biological reality stemming from the repeated exposure of a population to an identical/unchanging pathogen. Ideally, the theoretical study should be co-evolutionary, to allow the pathogen to also evolve. However, we understand that this introduces substantial complexity that might take focus away from the main aims of the paper. One suggestion that was positively discussed among the reviewers is to shorten the period of exposure.

(2) Building on Reviewer #2's comments about social information use, the manuscript should better acknowledge the fact that individuals can be flexible in their use of social vs. personal information, and that individual decisions to use different types of information can be context-dependent.

3) The authors will need to address the concerns about the handling of food items, the effect of infection status on foraging, and the lack of costs to reproduction. Reviewer #3 provides some useful suggestions, such as additional results, that could be presented to help alleviate the reader's concerns about these model design choices.

*Reviewer #1 (Recommendations for the authors):*

I have just a few comments or suggestions for the authors' considerations (these do not have to be followed, and there are few as the manuscript is overall very well written).

L36: Check the preprints to ensure that they haven't been published since drafting the manuscript.

L37: This paragraph doesn't really connect well to those around it.

L47: This is a great statement/justification.

L53: In Farine 2017 (https://besjournals.onlinelibrary.wiley.com/doi/full/10.1111/1365-2656.12764), I explored the question about the importance of the network formation process for understanding disease dynamics, and found that it is only important under certain conditions. Thus, the statement might be a bit more nuanced than presented here.

L53: It's unclear what sampling biases the authors mean. Are there some examples?

L68: It might be argued that this is not an eco-evo model.

L178: Movement is fixed, so it isn't really evolving.

L192: It's not totally clear what the purpose of extracting the networks is.

Discussion: The authors talk a bit about landscape productivity, but this isn't really explored much. Peng He (https://scholar.google.com/citations?user=ElcyoCIAAAAJ&hl=en) has several papers highlighting how the structure of the landscape might be important for social-based dynamics, including a simple model of disease spread in a 2021 paper in Oecologia. This might be of interest to the authors, or would at least be an interesting future direction for a model like the one they present here.

*Reviewer #2 (Recommendations for the authors):*

I congratulate the authors on their manuscript and I hope my comments are helpful in revising this interesting work.

There is an inconsistency in the manuscript on which outputs are explored -- which may be explained by the authors' search for simplicity. But I find it problematic to have limited access to the results. For example, by looking at Figure 2A, there is wide variation in individual responses to the "other" category -- which the authors later suggest is an artifact of mutations. If most of the cases are, as they state, related to agent tracking strategies, then it is worth extracting these data and analyzing them separately -- as they did for the two other categories.

It would be interesting to have a few examples of the pathogens the authors had in mind. Which pathogen would continuously affect a population, cause no death but drive considerable variation in social proximity in an asexual organism?

*Reviewer #3 (Recommendations for the authors):*

I enjoyed reading the manuscript and the authors generally did a nice job of presenting their rationale and findings. As I noted in my public review, I can envision many interesting questions that could be addressed using the approach developed here. My main concern relates to how reproduction was handled in the model and the consequences for which social movement strategies were therefore likely to be favored.

---

## [Author Response]

Overall, we have extended our model’s functionality while also fixing a small but important error in the initial code (v.1.1.0 on Github and Zenodo). We refer to this new version as v.1.2.0 (on Github and Zenodo).

**Fixing exploitation competition**

We have found and fixed a bug in the model that meant that exploitation competition was not implemented correctly. In the original model, individuals would first all scan their location for food items, and then select one for consumption (‘finding step’). Having all selected an item they would then actually forage in a randomised order (‘foraging step’). The original implementation relied on the assumption that individuals would be found at random locations on the landscape, and might only rarely select the same item as another individual, i.e. that exploitation competition would be rare. A check on item availability at the finding step was missing in the foraging step. This created a situation in which multiple individuals could select and start handling the same item within a single timestep. The item would only be truly unavailable in the next timestep. In combination with the clustering of food items and the handling time, this had the effect of dampening exploitation competition, which was expected to be a key feature of the model. This led to individuals clustering into a few (1 – 5) small, dense groups (our previous Fig. 3A).

This error has now been fixed, and the section below on *Model updates and corrections* explains in more detail how we have added tests to safeguard the intended ecological and evolutionary mechanisms against the possibility of this and other technical errors.

We would note that some of our results have changed as a result of fixing this error, but that the main evolutionary and ecological findings relating to a change in animal sociality do still hold. We previously reported that prior to pathogen introduction, individuals were indiscriminately social and ‘agent tracking’ - we now find that, as in our ‘global natal dispersal’ implementation, most individuals in the absence of a pathogen are ‘handler tracking’. This is in line with our findings from a model of the joint evolution of movement and competition (Gupte et al. 2021). After pathogen introduction however, the mixture of strategies is as we reported before, with a sharp transition away from sociality.

We also previously reported that there was a sharp drop in intake due to individuals not using social information - we now find that there is no change in intake (as in the ‘global dispersal’ model). However, there are still substantial changes in movement distance and social associations, as we previously reported. We also find that prior to pathogen introduction, the degree distribution is not right-skewed. We have corrected the text and figures to reflect these changes.

**Recovering earlier results**

In the earlier version of this manuscript, we reasoned that agent tracking was the commonest strategy before pathogen introduction because the time cost (and thus lost intake) of moving between food item clusters was greater than the cost of exploitation competition. Our reasoning would imply that increasing the costs of moving between patches would allow us to recover our earlier evolutionary results (initial agent tracking, with a sharp transition to handler tracking and agent avoidance).

We see two ways of increasing patch-switching costs: directly, by adding a small cost to movement, and indirectly, by altering the spatial structure of the landscape to have fewer, denser patches (keeping food items constant), such that moving between them takes more time. We explored both alternatives. Adding a cost of 0.05 for each unit of movement in our default implementation (R = 2, dE = 0.25, N clusters = 60, spread = 1.0), we were able to recover the evolutionary pattern we reported earlier. These results are not shown in the manuscript to avoid taking away from the main message.

On the other hand, encouraged by the reviewer's suggestion about the importance of spatial structure, we now discuss in the main text how two alternative spatial structures (uniform resource distribution, and more clustered distribution [N patches = 10, spread = 1.0]; see new **Fig. 10B**) affect the evolutionary outcomes before and after pathogen introduction. In brief, we show that under the more clustered distribution, we are able to recover our initial results. We hope this reassures reviewers that our earlier reasoning was correct.

**Additional model scenarios and implementations**

We now refer to our model’s main results as **scenario 1**, with pathogen introduction in each generation. In addition to our initial alternative implementations of this scenario (‘global dispersal’ and ‘percentage infection costs’), we have added two further implementations in line with the reviewers’ suggestions: (1) an implementation with a reproduction threshold, such that only individuals with a positive energy balance (intake - infection costs) can reproduce; and (2) implementations with different handling time durations, which is the main parameter controlling the persistence of social information.

We now also implement an additional **scenario 2**, in which there is a single pathogen introduction event, but in which the pathogen can be transmitted across generations. This allows us to address a suggestion to reduce the period of exposure to the pathogen, while also investigating the ecological and evolutionary causes and consequences of the pathogen persisting (or being eliminated) in the population.

Essential revisions:1) The reviewers were all concerned about the lack of biological reality stemming from the repeated exposure of a population to an identical/unchanging pathogen. Ideally, the theoretical study should be co-evolutionary, to allow the pathogen to also evolve. However, we understand that this introduces substantial complexity that might take focus away from the main aims of the paper. One suggestion that was positively discussed among the reviewers is to shorten the period of exposure.

We acknowledge that repeated exposure to an unchanging pathogen over thousands of generations is somewhat unlikely. We also agree with the editor that implementing both pathogen and host evolution could make the model much too complex to interpret. We have taken the reviewer suggestion to shorten the period of exposure on board, and implemented scenario 2 described above. This allows us to explore an additional question – under what conditions is the evolutionary transition away from sociality sufficient to eliminate the pathogen from the population, and how does sociality evolve once the pathogen is eliminated? We are now able to show how the infection cost and the probability of transmission from parents to offspring influence the evolution of sociality, and how these evolutionary dynamics in turn influence whether a pathogen can become established in a population, or whether it is eventually eliminated.

2) Building on Reviewer #2's comments about social information use, the manuscript should better acknowledge the fact that individuals can be flexible in their use of social vs. personal information, and that individual decisions to use different types of information can be context-dependent.

We appreciate the reviewers’ point here and have added the clarification that individuals can indeed be selective in their use of social and personal information; this in sections where we explain the movement process (L. 686 – 689), and in the Discussion (L. 598 – 602).

3) The authors will need to address the concerns about the handling of food items, the effect of infection status on foraging, and the lack of costs to reproduction. Reviewer #3 provides some useful suggestions, such as additional results, that could be presented to help alleviate the reader's concerns about these model design choices.

We appreciate the reviewers’ points regarding these mechanisms, and have taken them on board by showing the outcomes of two implementations that vary the handling time, and that introduce a threshold on reproduction. We also appreciate the point regarding the effect of infection on foraging, and feel that our ‘percentage costs’ implementation of scenario 1 addresses just this point. We have now rephrased how we refer to this implementation to make it clear that it can be interpreted as a reduction in foraging efficiency due to infection.

Reviewer #1 (Recommendations for the authors):I have just a few comments or suggestions for the authors' considerations (these do not have to be followed, and there are few as the manuscript is overall very well written).L36: Check the preprints to ensure that they haven't been published since drafting the manuscript.

We have updated all references to preprints to their versions published in journals.

L37: This paragraph doesn't really connect well to those around it.

The paragraph has been edited to make it clear that we seek to draw attention to the evolutionary consequences of pathogen introductions for hosts, rather than the more commonly detected and studied demographic effects (L. 55 – 62).

L47: This is a great statement/justification.

We thank the reviewer for the appreciation.

L53: In Farine 2017 (https://besjournals.onlinelibrary.wiley.com/doi/full/10.1111/1365-2656.12764), I explored the question about the importance of the network formation process for understanding disease dynamics, and found that it is only important under certain conditions. Thus, the statement might be a bit more nuanced than presented here.

We appreciate the point, and have changed this text (L. 73 – 79) to:

“Networks constructed from relatively low-resolution spatial relocation data (such as infrequent direct observations; see e.g. Albery et al. 2021), may be sensitive to the network formation process when seeking to understand the rapid spread of diseases, especially if transmission has a non-linear relationship with association strength (Farine 2017, White et al. 2017).”

L53: It's unclear what sampling biases the authors mean. Are there some examples?

Following from the previous sentence (L. 73 – 79):

“While high-resolution animal tracking could help construct more detailed networks (Nathan et al. 2022), such data may yet be sensitive to biases in animal tagging, such as when sociality is correlated with capture probability (see e.g. Carter et al. 2012).”

L68: It might be argued that this is not an eco-evo model.

Changed to say “…ecological and evolutionary consequences of…”

L178: Movement is fixed, so it isn't really evolving.

We have updated the line to clarify that it is movement decision-making that is evolved (L. 756 – 764).

L192: It's not totally clear what the purpose of extracting the networks is.

We have added text at lines 778 – 782 to make it clear that we did this to understand how changes in the individual-level preferences for social interactions could lead to emergent changes in the broader social structure of the population, and the consequences for pathogen transmission.

Discussion: The authors talk a bit about landscape productivity, but this isn't really explored much. Peng He (https://scholar.google.com/citations?user=ElcyoCIAAAAJ&hl=en) has several papers highlighting how the structure of the landscape might be important for social-based dynamics, including a simple model of disease spread in a 2021 paper in Oecologia. This might be of interest to the authors, or would at least be an interesting future direction for a model like the one they present here.

We appreciate that landscape spatial structure could influence social dynamics, and we have noted above how exploring the effect of spatial structure allowed us to recover our earlier results. We discuss this aspect in the main text (L. 408 – 445), with a figure showing strategy frequencies over 10 replicates (new Figure 10).

Reviewer #2 (Recommendations for the authors):I congratulate the authors on their manuscript and I hope my comments are helpful in revising this interesting work.There is an inconsistency in the manuscript on which outputs are explored -- which may be explained by the authors' search for simplicity. But I find it problematic to have limited access to the results. For example, by looking at Figure 2A, there is wide variation in individual responses to the "other" category -- which the authors later suggest is an artifact of mutations. If most of the cases are, as they state, related to agent tracking strategies, then it is worth extracting these data and analyzing them separately -- as they did for the two other categories.

We thank the reviewer for the kind words, and have taken the suggestion to revamp Figure 2 (new Figure 3, following the suggestion below). Figure 3A now clearly shows the three main strategies in our simulation. Furthermore, Figure 3 shows the raw and mean numbers of associations per overall movement for each of the three strategies. In contrast the previous version of this figure showed the average movement and average associations in each generation per strategy. We believe that this new visualisation (of the same underlying data) better illustrates the within- and between-strategy differences in the movement-association relationship.

It would be interesting to have a few examples of the pathogens the authors had in mind. Which pathogen would continuously affect a population, cause no death but drive considerable variation in social proximity in an asexual organism?

We appreciate the point that the exact circumstances of our model might not be the most common among empirical systems. However, we would emphasise that our model aims to be a conceptual exploration. We do believe that it could be adapted to fit more specific study systems by including host-pathogen trait co-evolution or by modelling sexual reproduction. For now, we have included a number of examples of the kinds of host-pathogen systems that our model could plausibly represent, such as foot-and-mouth disease in African buffalo, mange in North American wolves, or SARS-CoV-2 in North American deer (L. 107 – 118).

Reviewer #3 (Recommendations for the authors):I enjoyed reading the manuscript and the authors generally did a nice job of presenting their rationale and findings. As I noted in my public review, I can envision many interesting questions that could be addressed using the approach developed here. My main concern relates to how reproduction was handled in the model and the consequences for which social movement strategies were therefore likely to be favored.

We appreciate the reviewer’s interest in our work, and would point them to our implementation of the reproduction threshold and corresponding results.

**Model updates and corrections**

We have undertaken a series of corrections and updates to the simulation model code, and this follows directly from our finding of the error around the implementation of exploitation competition. These updates have been incorporated into version 1.2.0 of the model on Github, which is now referenced in the text. We have also referenced the model archived on Zenodo for better reproducibility. These updates are listed here.

**Conceptual changes**

A reorganisation of the scenario numbering, wherein scenario 1 refers to the default scenario of repeated introduction in each generation; scenario 2 refers to the additional scenario presented in the main text with only a single introduction event; and scenario 3 refers to an implementation in which the pathogen is introduced sporadically to the population.The addition of an option for a threshold on reproduction to be applied, which prevents individuals with a net energy balance < 0 from being considered as parents for the next generation. Implementing this option by setting ‘reprod_threshold = TRUE’, switches how fitness is handled at the end of each generation. First, a newly written function checks whether any individuals have a net energy > 0, and then, a second function filters for these individuals and stores their identities and their normalised energy values (0.0 – 1.0), and passes them to the weighted lottery to be drawn as parents. When no individuals have energy > 0, no parents can be drawn, and the simulation ends safely with a message, and stores the last generation’s trait information and adjacency matrix for potential analysis.

**Technical changes**

Addition of a C++ linting and checking workflow using two common linting and static code analysis tools. Adding this workflow helps to ensure the readability of the code and checks it for any common programming issues. The code is developed on Github with a master/develop workflow, and exploratory changes are not implemented until the checks in this workflow pass.Addition of MacOs testing for the overall R package ‘pathomove’ to ensure that the code runs on MacOs.Addition of a code test coverage workflow that checks whether the functions in the package are correctly tested. Current coverage is at 88%, as some functions relate to using an HPC cluster and cannot be tested.The addition of tests for the C++ code underlying the R function that runs the simulation. Specifically, the code tests:That individuals in a population are correctly intialised, with movement-decision making weights set to zero, and are placed within the bounds of the landscape;That reproduction and inheritance work correctly, with offspring assigned to the parent with the highest energy when only a single individual has had intake;That exploitation competition is correctly implemented, with only one of two individuals able to harvest a food item when it is initially available to both of them;That reproduction and inheritance work correctly when a threshold on energy prior to reproduction is applied;That individuals move as expected when they have a specific preference for an environmental cue;That vertical transmission of infection from parents to offspring works correctly;That position wrapping on a torus works correctly.Replacement of all C++ standard library random number generation functions with their {Rcpp} equivalents. This allows the simulation to be fully reproducible by passing a seed to the simulation through the main function. We chose this option so as to log the seed explicitly (as an R function, the seed could also have been set in R). We store the seeds for each simulation in the parameter files. This option does not work with multi-threading, and for full reproducibility multi-threading is turned off. Furthermore, dropping C++ standard library functions prevents package build errors related to using internal C++ random number engines. In cases where standard library functions were required, such as in shuffling the order of individuals, or selecting a random food item, we have implemented a wrapper around R’s random number generator as recommended by the {Rcpp} developers here: https://gallery.rcpp.org/articles/stl-random-shuffle/Individuals’ traits are now drawn from a normal distribution with a mean of 0.0 and a standard deviation that corresponds to the mutation size (0.01).Correction of the multithreaded movement step in which the first individual in the population (in randomised order) was not allowed to make movement or foraging decisions due to an indexing error in the setup of the multithreading step.The use of {Rcpp} numeric matrices to speed up the assignment of assessment errors used in the assignment of suitability scores in the movement step.Correction of the movement logging step so that decisions to remain at the current location are not logged as part of the distance moved.Correction to the wrapping function used to calculate the transformation from a coordinate that is beyond the bounds of the landscape to a coordinate that is on the diametrically opposite side, to correctly implement movement on a torus. This function relied on the use of ‘fmod’ from the standard library in C++, but is now fully manual and uses a simple formula instead.Corrections to the function to pick a food item from among nearby food items, such that the individual now picks a truly random item.Corrections to the foraging step such that individuals check whether their chosen item is actually available before harvesting it. This is a major fix and is described in detail in earlier sections of this Response.Correction to the function that normalises net energy. Previously, the function was coded to add small error terms (drawn from N(0.0, 0.001)) to the vector of normalised energies. This would result in some individuals potentially having negative energies after normalisation, especially if they had low energy (i.e., were normalised to close to 0.0). As a result their probability to reproduce would have been zero rather than the (very small) probability that would be suggested by our model description. The corrected function adds a small error term to individuals’ energy, and then normalises them. The previous implementation was a technical problem because the C++ standard library’s discrete distribution permits some, but not all, elements of a probability weights vector to have negative or zero values. In fact, when passed a vector, this object has an anomalous behaviour that is poorly documented: it internally normalises the vector and assigns a weight of zero to *weights whose sign is the minority after addition of their absolute values*. This means that if all but one individual of a population had negative energy, it would be the lone individual with positive energy that would be assigned a zero probability of being chosen as a parent. This is contrary to our intention. Using the {Rcpp} sample function guards against this behaviour as the function only accepts positive weights (including zero).Updates to the reproduction function to use {Rcpp} random number generation for the weighted lottery (using the ‘sample’ function), as well as to generate the new generation’s positions and potential new trait values after random mutations. These use the helper functions provided for C++ use: ‘rbinom’, ‘rnorm’, and ‘rcauchy’ which are similar to their R equivalents.Simplification of how data objects that record the traits in each generation of the simulation are created.Updates to how the randomised positions of food items are drawn to use {Rcpp} random number generation.Removal of excess, unused code.Change to the pathogen introduction function such that specifying a particular number of infections in each generation ensures that these infections are over and above any individuals that might have been infected by vertical transmission. This is primarily to cover the use case in which both vertical transmission as well as repeated pathogen introduction (‘scenario = 1’ or ‘scenario = 3’) are selected. The function ends if all individuals, by some chance, have been infected vertically.Change to the pathogen spread function to go through the individuals in random order. This change does allow individuals to be infected and also to transmit within the same timestep. This is a deliberate choice. Implementations of the model with transmission only possible after one timestep of infection have yielded the same results as the current implementation. Furthermore, changes to the spread function to log the infecting individual, to allow the construction of transmission chains.Removal of functions to classify the sources of infection and classify them, as this is logged in the data.Addition of a previously removed step that logs the movements of all individuals in the generation of pathogen introduction, as well as the final generation of the simulation.Better input checking and output printing when running the simulation.

**References**

Albery, G. F. et al. 2021. Multiple spatial behaviours govern social network positions in a wild ungulate. - Ecology Letters 24: 676–686.

Bastille-Rousseau, G. and Wittemyer, G. 2019. Leveraging multidimensional heterogeneity in resource selection to define movement tactics of animals. - Ecology Letters 22: 1417–1427.

Gupte, P. R. et al. 2021. The joint evolution of animal movement and competition strategies. - bioRxiv in press.

Lion, S. and Boots, M. 2010. Are parasites ‘“prudent”’ in space? - Ecology Letters 13: 1245–1255.

Lloyd-Smith, J. O. et al. 2005. Superspreading and the effect of individual variation on disease emergence. - Nature 438: 355–359.

Nathan, R. et al. 2008. A movement ecology paradigm for unifying organismal movement research. - PNAS 105: 19052–19059.

Pusceddu, M. et al. 2021. Honey bees increase social distancing when facing the ectoparasite varroa destructor. - Science Advances 7: eabj1398.

Sánchez, C. A. et al. 2022. A strategy to assess spillover risk of bat SARS-related coronaviruses in Southeast Asia. - Nat Commun 13: 4380.

Stroeymeyt, N. et al. 2018. Social network plasticity decreases disease transmission in a eusocial insect. - Science 362: 941–945.

Wilber, M. Q. et al. 2022. A model for leveraging animal movement to understand spatio-temporal disease dynamics. - Ecology Letters in press.

Wille, M. and Barr, I. G. 2022. Resurgence of avian influenza virus. - Science 376: 459–460.